# Impacts of Aerosols on Seasonal Precipitation and Snowpack in California Based on Convection-Permitting WRF-Chem Simulations

Longtao Wu[1], Yu Gu[2], Jonathan H. Jiang[1], Hui Su[1], Nanpeng Yu[3], Chun Zhao[4], Yun Qian[5], Bin Zhao[2], Kuo-Nan Liou[2], and Yong-Sang Choi[1,6]

[1]*Jet Propulsion Laboratory, California Institute of Technology, Pasadena, CA, USA.*

[2]*Joint Institute for Regional Earth System Science and Engineering and Department of Atmospheric and Oceanic Science, University of California, Los Angeles, CA, USA*

[3]*Department of Electrical and Computer Engineering, University of California, Riverside, Riverside, CA, USA*

[4]*School of Earth and Space Sciences, University of Science and Technology of China, Hefei, Anhui, China*

[5]*Atmospheric Sciences and Global Change Division, Pacific Northwest National Laboratory, Richland, WA, USA*

[6]*Department of Environmental Science and Engineering, Ewha Womans University, Seoul, South Korea*

(Submitted to ACP)

Highlights:

1. Aerosols warm the California mountain tops through aerosol-snow interaction by local dust but cools the lower elevation areas through aerosol-radiation interaction and aerosol-cloud interaction by transported and local anthropogenic aerosols.

2. Aerosols reduce precipitation and snowpack in California primarily through aerosol-cloud interaction by transported and local anthropogenic aerosols and aerosol-snow interaction by local dust.

3. Aerosols cause early snowmelt at mountain tops through aerosol-snow interaction by local dust, and hence modify the seasonal cycle of surface runoff.

**Abstract**

A version of the WRF-Chem model with fully coupled aerosol-meteorology-snowpack is employed to investigate the impacts of various aerosol sources on precipitation and snowpack in California. In particular, the impacts of locally emitted anthropogenic and dust aerosols, and aerosols transported from outside California are studied. We differentiate three pathways of aerosol effects including aerosol-radiation interaction (ARI), aerosol-snow interaction (ASI), and aerosol-cloud interaction (ACI). The convection-permitting model simulations show that precipitation, snow water equivalent (SWE), and surface air temperature averaged over the whole domain (34-42°N, 117-124°W, not including ocean points) are reduced when aerosols are included, therefore reducing large biases of these variables due to the absence of aerosol effects in the model. Aerosols affect California water resources through the warming of mountain tops and the reduction of precipitation; however, different aerosol sources play different roles in changing surface temperature, precipitation and snowpack in California by means of various weights of the three pathways. ARI by all aerosols mainly cools the surface, leading to slightly increased SWE over the mountains. Locally emitted dust aerosols warm the surface of mountain tops through ASI, in which the reduced snow albedo associated with dusty snow leads to more surface absorption of solar radiation and reduced SWE. Transported aerosols and local anthropogenic aerosols play a dominant role in increasing non-precipitating clouds but reducing precipitation through ACI, leading to reduced SWE and runoff over the Sierra Nevada, as well as the warming of mountain tops associated with decreased SWE and hence lower surface albedo. The average changes in surface temperature from October 2012 to June 2013 are about −0.19 K and 0.22 K for the whole domain and over mountain tops, respectively. Overall, the averaged reduction during October to June is about 7% for precipitation, 3% for SWE, and 7% for surface runoff for the whole domain,

while the corresponding numbers are 12%, 10%, and 10% for the mountain tops. The reduction in

SWE is more significant in a dry year, with 9% for the whole domain and 16% for the mountain

tops. The maximum reduction of ~20% in precipitation occurs in May associated with the

maximum of aerosol loadings, leading to the largest decrease in SWE and surface runoff over that

period. It is also found that dust aerosols could cause early snowmelt at the mountain tops and

reduced surface runoff after April.

## 1.    Introduction

Water resources in California are derived predominantly from precipitation (mostly during

the winter time) and storage in the snowpack in the Sierra Nevada. Snowpack provides about one-

third of the water used by California's cities and farms. The fresh water stored in the snowpack

gradually releases through runoff into river flows during the warm and dry season. The amount

and timing of snowmelt are critical factors in determining water resources in this region. It is

important to understand the factors influencing precipitation and snowpack on seasonal timescale

for water management and hydropower operation.

The 2012-2014 California drought has been attributed to both warming and anomalously low

precipitation (Griffin and Anchukaitis, 2014). Previous studies suggested that warming trends are

amplified in mountains compared to lowlands (Pepin et al., 2015). The amplified warming in

mountain areas, also referred to as elevation-dependent warming, is generally attributed to a few

important processes (Pepin et al., 2015), such as water vapor changes and latent heat release,

surface water vapor changes, radiative flux changes associated with three-dimensional rugged

topography (Gu et al., 2012a; Liou et al., 2013; Lee et al., 2015; Zhao et al., 2016), and snow-

albedo feedback (Leung et al., 2004). A review and assessment of the mechanisms contributing to
an enhanced warming over mountain areas is given in Pepin et al. (2015).
In addition to the warming effects of greenhouse gases, aerosols may have substantial
impacts on water resources in California. Recent observational and numerical modeling studies
have shown that aerosol pollutants can substantially change precipitation and snowpack in
California (e.g., Rosenfeld et al., 2008a; Qian et al., 2009a; Hadley et al., 2010; Ault et al., 2011;
Creamean et al., 2013, 2015; Fan et al., 2014; Oaida et al., 2015). Lee and Liou (2012) illustrated
that approximately 26% of snow albedo reduction from March to April over the Sierra Nevada is
caused by an increase in aerosol optical depth (AOD).
In California, aerosols can be generated locally or transported from remote sources. Among
local aerosol types, dust comprises a significant fraction over California (Wu et al., 2017). Based
on a four-month, high intensity record of size-segregated particulate matter (PM) samples collected
from a high elevation site, Vicars and Sickman (2011) found that the mass concentration of coarse
atmospheric PM in the southern Sierra Nevada, California, was dominated by contribution from
dust (50-80%) throughout the study period. Dust aerosols can exert important impact on radiative
forcing and regional climate in California through its interaction with radiation (e.g., Zhao et al.,
2013a) as well as its role as cloud condensations nuclei for cloud formation (e.g., Fan et al., 2014).
Anthropogenic aerosols are geographically distributed because of localized emission sources, the
short atmospheric residence time, and regional topography. With valleys and surround mountain
barriers, dispersion of air pollutants is more difficult for locally emitted anthropogenic air
pollution. The anthropogenic aerosols can cause changes in atmospheric circulation and regional
climate especially where the aerosol concentrations are high and the synoptic atmospheric systems
are not prominent (e.g., Qian et al., 2003; Fast et al., 2006; Rosenfeld et al., 2008a; Zhao et al.,
2013a).

Besides the local aerosol sources, the atmospheric transport of aerosol pollutants from the

Asian continent (e.g., Jiang et al., 2007; Wang et al., 2015; Hu et al., 2016) is also a significant
contributor to aerosol loading throughout the Pacific basin. Asian aerosols can reach relatively
high concentrations above the marine boundary layer in the western US, representing as much as
85% of the total atmospheric burden of PM at some sites (VanCuren, 2003). Trans-Pacific dust
transport has been found to be particularly relevant in high-elevation regions such as the Sierra
Nevada, which typically represents free-tropospheric conditions due to the limited transport of
lowland air pollutants and predominance of upper air subsidence (VanCuren et al., 2005).
Observations from the CalWater campaign demonstrated that dust and biological aerosols
transported from northern Asia and the Sahara were present in glaciated high-altitude clouds in the
Sierra Nevada coincident with elevated ice nuclei (IN) particle concentrations and ice-induced
precipitation (Ault et al., 2011; Creamean et al., 2013).

Aerosols can influence precipitation, snowpack and regional climate through three pathways:

(1) aerosol-radiation interaction (ARI, also known as aerosol direct effect), which can warm the
atmosphere but cool the surface, resulting in changes in thermodynamic environment for cloud
and precipitation and the delay of the snowmelt (Charlson et al., 1992; Kiehl and Briegleb, 1993;
Hansen et al., 1997; Koren et al., 2004; Gu et al., 2006, 2016, 2017); (2) aerosol-cloud interaction
(ACI, also known as aerosol indirect effect), which is related to aerosols serving as cloud
condensation nuclei (CCN) and IN. By changing the size distribution of cloud droplets and ice
particles, aerosol may affect cloud microphysics, radiative properties and precipitation efficiency,
thus affect the atmospheric hydrological cycle and energy balance (Twomey, 1977; Jiang and

Feingold, 2006; Rosenfeld et al., 2008b; Qian et al., 2009b; Gu et al., 2012b); (3) aerosol-snow

interaction (ASI). When aerosols (mainly absorbing aerosols, such as dust and black carbon) are

deposited on snowpack, they can reduce snow albedo and affect snowmelt (Warren and Wiscombe,

1985; Jacobson, 2004; Flanner et al., 2007; Qian et al., 2011, 2015; Zhao et al., 2014). Numerical

experiments have shown that ARI reduces the surface downward radiation fluxes, cools the surface

and warms the atmosphere over California (Kim et al., 2006; Zhao et al., 2013a), which could

subsequently impact clouds, precipitation and snowpack. In a 2-D simulation, Lynn et al. (2007)

shows that ACI decreases orographic precipitation by 30% over the length of the mountain slope.

Fan et al. (2014) showed that ACI increases the accumulated precipitation of an Atmospheric River

event by 10-20% from the Central Valley to the Sierra Nevada due to a ~40% increase in snow

formation. Snow impurities (ASI) increase ground temperature, decrease snow water, shorten

snow duration and cause earlier runoff (Jacobson, 2004; Painter et al., 2007, 2010; Qian et al.,

2009a; Waliser et al., 2011; Oaida et al., 2015).

Although recent studies showed that aerosols can substantially influence precipitation and

snowpack in California, they focused only on one of the aerosol sources or on a single event or

one pathway. A complete account of the aerosol impacts from different sources through three

pathways on regional climate in California has not been presented yet. The objective of this study

is to investigate the impacts of various aerosol sources on seasonal precipitation and snowpack in

California. A fully coupled high-resolution aerosol-meteorology-snowpack model will be used.

We will distinguish and quantify the impacts of aerosols from local emissions and transport, and

the roles of different prevailing aerosol types in California, particularly dust and anthropogenic

aerosols. In Section 2, we describe the WRF-Chem model employed and experiments designed to

understand the impact of aerosols on precipitation and snowpack in California. Results from model
simulations are discussed in Section 3. Concluding remarks are given in Section 4.

**2.      Model Description and Experiment Design**
This study uses a version of the Weather Research and Forecasting (WRF) model with
chemistry (WRF-Chem; Grell et al., 2005) improved by the University of Science and Technology
of China (USTC) based on the public-released version 3.5.1 (Zhao et al., 2014). ASI is
implemented in this WRF-Chem version by considering aerosol deposition on snowpack and the
subsequent radiative impacts through the SNow, ICe, and Aerosol Radiative (SNICAR) model
(Zhao et al., 2014). The SNICAR model is a multilayer model that accounts for vertically
heterogeneous snow properties and heating and influence of the ground underlying snow (Flanner
and Zender, 2005; Flanner et al., 2007, 2009, 2012). The SNICAR model uses the theory from
Wiscombe and Warren (1980) and the two-stream, multilayer radiative approximation of Toon et
al. (1989). SNICAR simulates snow surface albedo as well as the radiative absorption within each
snow layer. It can also simulate aerosol content and radiative effect in snow, and was first used to
study the aerosol heating and snow aging in a global climate model by Flanner et al. (2007).
Simulated change of snow albedo by SNICAR for a given black carbon concentration in snow has
been validated with recent laboratory and field measurements (Brandt et al., 2011; Hadley and
Kirchstetter, 2012). More detailed description of the SNICAR model can be found in Flanner and
Zender (2005) and Flanner et al. (2007, 2012).
The MOSAIC (Model for Simulating Aerosol Interactions and Chemistry) aerosol model
(Zaveri et al., 2008) with the CBM-Z (carbon bond mechanism) photochemical mechanism (Zaveri
and Peters, 1999) is used and coupled with the SNICAR model. The MOSAIC aerosol scheme
uses the sectional approach to represent aerosol size distributions with a number of discrete size
bins, either four or eight bins in the current version of WRF-Chem (Fast et al., 2006). In this study,
aerosol particles are partitioned into four-sectional bins with dry diameter within 0.039-0.156 μm,
0.156-0.625 μm, 0.625-2.5 μm, and 2.5-10.0 μm. The 4-bin approach has been examined in dust
simulations and proved to reasonably produce dust mass loading and AOD compared with the 8-
bin approach (Zhao et al., 2013b). All major aerosol components including sulfate, nitrate,
ammonium, black carbon, organic matter, sea salt, and mineral dust are simulated in the model.
The MOSAIC aerosol scheme includes physical and chemical processes of nucleation,
condensation, coagulation, aqueous phase chemistry, and water uptake by aerosols. Dry deposition
of aerosol mass and number is simulated following the approach of Binkowski and Shankar (1995),
which includes both particle diffusion and gravitational effects. Wet removal of aerosols by grid
resolved stratiform clouds/precipitation includes in-cloud removal (rainout) and below-cloud
removal (washout) by impaction and interception, following Easter et al. (2004) and Chapman et
al. (2009). In this study, cloud-ice-borne aerosols are not explicitly treated in the model but the
removal of aerosols by the droplet freezing process is considered. Aerosol optical properties such
as extinction, single scattering albedo (SSA), and asymmetry factor for scattering are computed as
a function of wavelength for each model grid box. Aerosols are assumed internally mixed in each
bin, i.e., a complex refractive index is calculated by volume averaging for each bin for each
chemical constituent of aerosols (Barnard et al., 2010; Zhao et al., 2013a). The Optical Properties
of Aerosols and Clouds (OPAC) data set (Hess et al., 1998) is used for the shortwave (SW) and
longwave (LW) refractive indices of aerosols, except that a constant value of $1.53+0.003i$ is used
for the SW refractive index of dust following Zhao et al. (2010, 2011). A detailed description of
the computation of aerosol optical properties in WRF-Chem can be found in Fast et al. (2006) and
Barnard et al. (2010).

ARI is included in the radiation scheme as implemented by Zhao et al. (2011). The optical

properties and direct radiative forcing of individual aerosol species in the atmosphere are
diagnosed following the methodology described in Zhao et al. (2013a). The activation and re-
suspension between dry aerosols and cloud droplets are included in the model as shown in
Gustafson et al. (2007). By linking simulated cloud droplet number with shortwave radiation and
microphysics schemes, ACI is effectively simulated in the model (Chapman et al., 2009).

The model setups (Table 1), including the physical schemes used, follow Wu et al. (2017),

which showed that the model simulations reasonably captured the distribution and variation of
aerosols in the San Joaquin Valley. The model domain covers the Western US centered at 38°N
and 121°W, as shown in Fig. 1. The horizontal resolution is 4 km × 4 km together with a vertical
resolution of 40 model levels. Model integrations with a time step of 20 seconds have been
performed for 10 months (with the first month used for the model spin-up) starting on September
1, 2012, at 00:00UTC till the end of June 2013 to cover the major precipitation and snow seasons.
To test the robustness of the results, simulations are also conducted for year 2013-2014, and similar
results are found. In the following section, our analysis focuses on year 2012-2013, while
quantitative information of the aerosol impacts for year 2013-2014 is provided for comparison.

Note that convective processes are resolved in the 4 km simulations. One important subgrid

process in climate models is the representation of deep convection. Parameterizing deep
convection is challenging and the use of convection parameterization schemes leads to common
errors such as misrepresentation of the diurnal cycle of convective precipitation (e.g., Dai et al.,
1999; Brockhaus et al., 2008), underestimation of dry days (e.g., Bergetal., 2013) and precipitation
intensity (e.g., Prein et al., 2013; Fosser et al., 2014; Ban et al., 2014), and overestimation of low-
precipitation frequency (e.g., Bergetal., 2013). Although recently developed parameterization
schemes lead to improvements in the simulation of precipitation intensity (Donner et al., 2011),
intraseasonal variability (Benedict et al., 2013), and diurnal cycles (Bechtold et al., 2014), a
promising remedy to the error-prone model simulations using convective parameterizations is the
use of convection-permitting horizontal resolution with grid spacing of about 4 km or less (e.g.,
Satoh et al., 2008; Prein et al., 2013; Ban et al., 2014). Advances in high-performance computing
allowed refinement of the model grids well below 10 km. At these scales, convection
parameterization schemes may be switched off as deep convection starts to be resolved explicitly
(e.g., Weisman et al., 1997). According to Prein et al. (2014), it seems prudent to use horizontal
grid spacing of 4 km or less for convection-permitting model simulations. The 4 km simulation
can also represent topography and inhomogeneous distribution of anthropogenic emission and
precipitation better, leading to a better representation of aerosol distribution comparing to the 20
km simulation (Wu et al., 2017).

Since the model explicitly considers different sources and types of aerosols and contains the

physical processes to represent various aerosol effects (ARI, ASI, and ACI), it is useful to
decompose the aerosol effects based on aerosol sources/types and pathways. Note that the overall
aerosols effects are not a simple sum of different aerosol sources/types, nor a linear combination
of the ARI, ASI, and ACI effects. Differences between various simulations, however, help to
identify the effect of a single source or pathway and the decomposition approach is a common
practice in the experiment design of modeling studies. To examine the overall aerosol effects and
the roles of locally generated and transported aerosols, the following five experiments have been
designed (Table 2):
1) CTRL: This is the control experiment with all aerosol emissions and transports included
in the simulation.
2) NoLocDust: This experiment is performed without any local dust emission. Differences
between the CTRL and NoLocDust experiments illustrate the effect of dust aerosols locally
emitted.
3) NoLocAnth: This experiment is similar to NoLocDust, except that emissions of local
anthropogenic aerosols are turned off. Comparison between CTRL and this experiment will
elucidate the effect of local anthropogenic aerosols.
4) NoTran: The initial and boundary chemical conditions in the CTRL simulation are taken
from the global Model for Ozone and Related Chemical Tracers, version 4 (MOZART-4; Emmons
et al., 2010). The chemical species transported into the model domain include organic carbon,
black carbon, sulfate, nitrate, ammonium, sea salt, dust, etc.. In the NoTran experiment, aerosols
transport from outside the model domain, including those from East Asia and other regions, are
not considered by setting the lateral boundary conditions for aerosols to zero. Differences between
CTRL and NoTran will show the effect of transported aerosols.
5) CLEAN: This experiment is performed without any local aerosol emissions or transport
from outside the model domain while all the transported chemical species are kept, and therefore
represents a scenario of clean condition. Aerosols are low in the simulation, but not zero, possibly
due to aerosol chemistry. The CCN concentration at supersaturation of 0.1% is on the order of 10
$cm^{-3}$ at most time of the CLEAN simulation. The distribution of liquid water path and ice water
path in the CLEAN simulation is also similar to that in the CTRL simulation, with differences in
magnitude. Differences between the CTRL and CLEAN experiments would illustrate the effects
of all primary aerosol types, including those locally emitted and transported from outside the
domain.

In order to distinguish the pathways through which the aerosols influence the precipitation

and snowpack, we also conducted a few other experiments (Table 3):

6) NARI: This experiment is similar to the CTRL run, except that ARI is not included.

Comparison between CTRL and this experiment will elucidate the effect of ARI.

7) NASI: This experiment is similar to the CTRL run, except that ASI is not included.

Comparison between CTRL and this experiment will show the effect of ASI.

8) NARS: This experiment is similar to the CTRL run, except that both ARI and ASI are not

included. By comparing this experiment and CLEAN, the effect due to ACI can be examined.

**3.     Model Simulation Results**
**3.1   Validation of Model Results**

Since our focus is on the changes in precipitation and snowpack due to aerosol effects, we

first show the spatial distribution of averaged results over the period from October 2012 to June
2013 when snow normally presents over the Sierra Nevada. Figure 2 illustrates a few important
and relevant variables that the model simulates in the CTRL experiment, including liquid water
path (LWP), ice water path (IWP), precipitation, snow water equivalent (SWE), and temperature
at two meters (T2) above the ground. SWE is a common snowpack measurement. It is the amount
of water contained within the snowpack and can be regarded as the depth of water over unit flat
surface that would theoretically result if the entire snowpack melted instantaneously. Here, the
model simulated SWE is the mean value of the accumulated SWE from 3-hourly model outputs.
It is shown that clouds (Fig. 2a and 2b), precipitation (Fig. 2c), snowpack (Fig. 2d), and surface
runoff (Fig. S1) mostly occur over the Sierra Nevada and Klamath Mountains in northern
California. For temperature (Fig. 2e), the central valley area appears to be relatively warm with
two maxima over the northern and southern part of the central valley, respectively, while colder
temperatures are found over the mountain ranges. The model-simulated precipitation is compared
with corresponding observations from the Parameter elevation Regression on Independent Slopes
Model (PRISM, 2004) gridded data product at 4 km resolution (Fig. 2f). Note that the precipitation
rate in comparison here is for total precipitation, including rainfall and ice-phase particles.
Compared to the PRISM observations, the model successfully captures the precipitation pattern,
including the locations of the major precipitation centers, but slightly overestimates the magnitude
over the Sierra Nevada.

In order to validate the simulated seasonal variations, the monthly mean model simulated

precipitation and T2 are compared with observations (Fig. 3a and 3c). Model data are sampled
onto observational sites before the comparison is conducted. For precipitation observations,
besides the PRISM product, we also employ the Climate Prediction Center (CPC) Unified Gauge-
Based Analysis of Daily Precipitation product (Chen et al., 2008) at $0.25° \times 0.25°$ resolution and
the gauge measurements from Department of Water Resources (DWR). Observed air temperature
is obtained from the California Irrigation Management Information System (Snyder, 1984). For
SWE, daily accumulated SWE simulations are compared with measurements collected at Snow
Telemetry (SNOTEL) stations. SNOTEL SWE is measured using a snow pillow sensor and biases
in SWE measurement could occur when temperature differences between surrounding ground
cover and the pillow sensor create uneven distribution of snow (Meyer et al., 2012). Both under-
and over-estimation could happen depending on the snowmelt conditions and the snow density
rate of change (Serreze et al., 1999; Serreze et al., 2001; Johnson and Marks, 2004).
It is shown that the model captures the maximum precipitation in December, with the
magnitude falling between the observations from CPC and PRISM/DWR during winter, which is
the major rainy season in California (Fig. 3a). In the relative dry months from February to June,
the simulated precipitation has similar magnitude to the observations, with slightly overestimation
or underestimation in different months. For SWE, the model simulations represent seasonal
variations of SWE with the maximum between March and April (Fig. 3b), but the model
overestimates SWE amount comparing to SNOTEL. While the model overestimates the surface
temperature in magnitude, it captures the seasonal variations well, including the highest/lowest
temperature in July/January, respectively (Fig. 3c).
The simulated aerosols over California using this model have been validated extensively in
Wu et al. (2017) by comparing to observations, such as MISR (Multiangle Imaging
Spectroradiometer) and AERONET (Aerosol Robotic Network) AOD, CALIPSO (Cloud-Aerosol
Lidar and Infrared pathfinder Satellite Observation) aerosol extinction, IMPROVE (Interagency
Monitoring of Protected Visual Environments) and EPA CSN (National Chemical Speciation
Network operated by Environmental Protection Agency) aerosol speciation. It has been shown
than the model simulation used in this study reasonably captures the distribution and seasonal
variation in aerosols during the cold season from October to March. The simulation of aerosols in
the warm season from April to September (especially from July to September) has larger low
biases than in the cold season, mainly due to poor simulations of dust emission and vertical mixing.
Because the precipitation and snow mainly occur in October-June, we focus on the simulations
from October to June with relative good performance on aerosol simulations in this study.
Here, we present the distributions of AOD averaged over October 2012 to June 2013 for the
MISR (Diner et al., 1998) observation and all aerosols in the CTRL simulation, together with
locally emitted aerosols and those transported from outside the model domain, derived from the
difference between the CTRL simulation and the corresponding experiment (NoLocAnth,
NoLocDust and NoTran), respectively, to facilitate the understanding of the aerosol effects in
different regions and from different sources (Fig. 4). It is shown that the model simulation well
captures the spatial distribution of AOD in California, including the maximum over the southern
part of the valley area and larger AODs over the lower lands to the southeast of the Sierra Nevada
(Fig. 4a and 4b). Note that the smoother contour in MISR is due to the coarser horizontal resolution
(0.5°) of the MISR data. The distribution of the locally emitted anthropogenic aerosols (Fig. 4c),
which are mostly located over the central valley associated with the emissions from local industries
and farms, presents a similar pattern to the total AOD and substantially contributes to the maxima
AOD over the region. Local dust aerosols mainly reside over the lower lands to the southeast of
the Sierra Nevada while substantial amounts are also seen over the central valley (Fig. 4d).
Transported aerosols are carried into the domain by atmospheric circulation and widely distributed,
with more over the central valley due to the trapping of aerosols by the surrounding mountains
(Fig. 4e).

Since the observations on aerosol-in-snow concentrations are rather limited both spatially

and temporally, it's very difficult to conduct direct comparisons with model simulations. Here we
evaluate the model simulations of snow albedo which is directly affected by the ASI (Fig. S2). The
model simulated snow albedo is compared with the product from NASA Land Data Assimilation
Systems (NLDAS; Sheffield et al., 2003) Mosaic (MOS). It is shown that model simulation
provides rather reasonable estimate of the snow albedo when ASI is included. Overall, the WRF-
Chem model that we employ in this study is a reliable tool for examining the impact of aerosols
on the seasonal variations of precipitation and snowpack in California, especially over the Sierra
Nevada.

**3.2   Aerosol Effects on Precipitation and Snowpack**
The overall aerosol effects, from all aerosol types and sources (including locally emitted and
transported) through the three pathways (ARI, ASI, and ACI), can be examined from the
differences between the experiments CTRL and CLEAN. The two-tailed Student's t test, in which
deviations of the estimated parameter in either direction are considered theoretically possible, is
applied to the 3-hourly data for each experiment in this study to measure the statistical significance
of the simulations. Figure 5 shows the differences averaged over October 2012 to June 2013 in
precipitation, SWE, and T2, where the dots represent differences of the 3-hourly data being
statistically significant at above 90% level. Due to the aerosol effects, temperature decreases over
the central valley, where most aerosols are located, while significant warming occurs over the
mountain tops (Fig. 5c). Precipitation decreases over the Sierra Nevada (Fig. 5a), consequently
leading to decreased SWE (Fig. 5b).
In order to understand how the aerosols affect these important variables, we examine the
effects of ARI, ASI, and ACI separately. In the following figures (Fig. 6 to Fig. 12), the differences
are statistically significant at 70% level. It is seen that the major effect of ARI is to decrease the
surface temperature over the whole domain through the scattering and absorption of solar radiation,
with the maxima over the central valley where the aerosols are mostly located, contributing to the
surface cooling caused by the total aerosols effects in that region (Fig. 6c). The ARI induced
surface cooling over the Sierra Nevada, although not as strong as over the central valley, leads to
reduced snowmelt and hence slight increase in SWE, opposite to the overall aerosol effect on SWE
(Fig. 6b). The effect of ARI on rainfall is not very significant (Fig. 6a). The main effect of ASI is
to increase the temperature (Fig. 7c) over the snowy area of the Sierra Nevada through the
reduction of snow albedo (Fig. 7d) and hence more absorption of solar radiation at the surface,
contributing to the reduced SWE over the Sierra Nevada (Fig. 7b). The effect of ASI on
precipitation is also minimal.

Figure 8 shows the effect of aerosols on clouds through ACI. When more aerosols are present

in the atmosphere, more CCN are available for the formation of clouds with smaller cloud droplets.
As a result, more non-precipitating clouds are produced when aerosol are included in the model.
The enhanced LWP (Fig. 8a) is primarily produced by the ACI effect (Fig. 8c). There are no
significant changes in IWP (including ice, snow, and graupel) because the aerosol effect on ice
cloud formation is not explicitly treated in the model. The ACI effect leads to reduced precipitation
and less SWE over the mountains (Figs. 9a and 9b). Temperature decreases over the valley due to
more clouds formed associated with the ACI effect. The increase in temperature over the mountain
areas (Fig. 9c) is caused by the reduced snow amount, which results in weaker surface albedo (Fig.
9d) and enhanced solar absorption at the surface and overwhelms the decrease of temperature
which may be caused by increased clouds.

Overall, aerosols affect surface temperature, precipitation, and snowpack in California

through the three pathways. ACI plays a dominant role in increasing cloud water but reducing
precipitation, leading to reduced SWE and surface runoff (Fig. S3) over the Sierra Nevada. ASI
also reduces SWE due to the smaller snow albedo associated with dirty snow, leading to more
surface absorption and snowmelt. ARI, on the other hand, slightly increases SWE through the
cooling of the surface. For surface temperature, ARI and ACI contribute together to the cooling of
the valley area, while ACI and ASI significantly warm the surface over the mountain tops. Note
that for the ASI effect, warming of the snow cover area through aerosol induced snow-albedo
feedback is the cause for the reduced SWE. For the ACI effect, however, warming over the
mountain region is a result from the reduced SWE which can also induce snow-albedo feedback
and result in smaller surface albedo and more surface absorption of solar radiation.
Next, we examine the roles of local anthropogenic aerosols and local dust as well as
transported aerosols. The effect of local anthropogenic aerosols can be discovered from the
differences between CTRL and NoLocAnth. It is shown that local anthropogenic aerosols slightly
suppress precipitation (Fig. 10a) via ACI, leading to a reduced SWE (Fig. 10b) and a warming
over the mountain tops (Fig. 10c). The cooling of the valley area, where locally emitted
anthropogenic aerosols are mostly located (Fig. 4b), is associated with both the ARI effect and
more non-precipitating clouds produced through ACI. Dust aerosols emitted from local sources
mainly warm the surface through the reduction of snow albedo (ASI, Fig. 11c), consequently
enhancing the snowmelt and leading to the reduced SWE (Fig. 11b). Local dust aerosols have no
significant effect on precipitation (Fig. 11a).
Note that the effects of local anthropogenic and dust aerosols do not seem to be able to
explain the total effects of aerosols as seen in Fig. 5, raising the question whether the transported
aerosols play an important role in the precipitation and snowpack over the Sierra Nevada. Figure
12 illustrates the impact of aerosols transported from outside the model domain. It is shown that
transported aerosols reduce the precipitation through ACI (Fig. 12a), which exceeds the ARI effect
and leads to decreased SWE and increased temperature over the southern part of the Sierra Nevada
(Fig. 12b and 12c). Over the central valley, as well as over the northern part of the Sierra Nevada,
temperature decreases (Fig. 12c) due to the relatively larger ARI effect of the transported aerosols
compared to the ACI effect, resulting in less snowmelt and increased SWE over that region (Fig.
12b).
The overall changes induced by aerosols for surface temperature (K) and precipitation, SWE,
and surface runoff in percentage averaged over October to June are given in Table 4 for the whole
domain (34-42 °N, 117-124 °W, not including ocean points), mountain tops (elevation $\geq$ 2.5 km),
and lower elevations (elevation < 2.5 km).  For the whole domain in year 2012-2013, temperature
is cooled by 0.19 K due to aerosol ARI (−0.14 K), as well as ACI (−0.06 K) mainly associated
with transported aerosols (−0.17 K), accompanied by reduction in precipitation, SWE, and surface
runoff of about 7%, 3%, and 7%, respectively. Reduction in precipitation is mainly caused by ACI
(−6.26%) associated with transported (−2.97%) and local anthropogenic (−1.02%) aerosols. For
SWE, reduction is attributed to ACI (−2.67%) and ASI (−1.96%), while ARI contributes to an
increase (1.88%). Surface runoff is defined as water from precipitation, snowmelt, or other sources
that flows over the land surface, and is a major component of the hydrological cycle. Overall
changes in surface runoff are similar to those in precipitation, accompanied by contributions from
changes in snowmelt. For the mountain tops, warming of 0.22 K is found attributed to ASI (0.12
K) and ACI (0.17 K) associated with local dust and anthropogenic aerosols, respectively, with 10%
or more reduction in precipitation, snowpack, and surface runoff. Therefore, aerosols may
contribute to California drought through both the warming of mountain tops and anomalously low
precipitation over the whole area. For the lower elevations, the domain averaged changes are
similar to those for the whole domain, except for SWE which slightly increases by 0.42% due to
ARI (2.43%) with main contribution from transported aerosols (4.01%).
The simulations for year 2013-2014 are consistent with those in year 2012-2013 (Table 4).
For the whole domain in year 2013-2014, temperature is cooled by 0.21 K due to aerosols,
accompanied by reduction in precipitation, SWE, and surface runoff of about 6%, 9%, and 5%,
respectively. Aerosol impacts on SWE is more significant in year 2013-2014 (−8.88%) than in
year 2012-2013 (−3.17%), possibly due to less precipitation and SWE in year 2013-2014 than year
2012-2013 (not shown). The changes of SWE for year 2013-2014 are −15.57% for the mountain
tops and 2.66% for the lower elevations. The relative change of surface runoff at the mountain tops
in year 2013-2014 is smaller than year 2012-2013 because the mean surface runoff in year 2013-
2014 (0.33 mm day$^{-1}$) is larger than that in year 2012-2013 (0.27 mm day$^{-1}$), possibly contributed
by less SWE and faster snowmelt at the mountain tops in year 2013-2014. The corresponding
changes in evapotranspiration are −0.12% in year 2012-2013 and −1.20% in year 2013-2014,
respectively, which also contributes to the relatively smaller change of surface runoff in year 2013-
2014 at the mountain tops.

**3.3   Seasonal Variations of Aerosol Effects**
Figure 13 depicts the monthly mean AOD for total aerosols (brown solid), local
anthropocentric aerosols (green dashed), local dust (blue dashed), and transported aerosols (red
dashed) averaged over the whole domain, mountain tops, and lower elevation area from October
2012 to June 2013. It is seen that transported aerosols contribute to about two-thirds of the total
AOD. The total AOD has two maxima, one in December and one in May, mainly associated with
the seasonal variations of transported aerosols and local dust aerosols. Local dust AOD starts to
increase in March and reaches a maximum around May, while transported aerosol AOD peaks in
April (Fig. 13a). The seasonal variations of AOD over the mountain tops and lower elevations are
similar to those of the whole domain (Figs. 13b and 13c).

The monthly mean differences in precipitation due to the total aerosols (brown solid), ARI

(green solid), ASI (blue solid), ACI (red solid), local anthropocentric aerosols (green dashed), local
dust (blue dashed), and transported aerosols (red dashed) are shown in Fig. 14. Reduced
precipitation is seen over the whole domain, with the most contribution from transported aerosols,
followed by local anthropogenic aerosols, both of which play roles in precipitation changes
through ACI as previously shown. ARI, ASI, or locally emitted dust aerosols do not seem to play
an important role in the monthly mean precipitation changes (Fig. 14a). Two maxima of aerosol
effects are found: one in December when it is the rainy season of the California (Fig. 3a) and at
the same time relatively larger AOD presents over this region (Fig. 13a); the other peak reduction
in precipitation due to the aerosol effects is found in May with a value of about 0.2 mm day$^{-1}$ (Fig.
13a), probably associated with the maximum aerosols (Fig. 13a) and also the orographic
precipitation over the mountain region during that time period (Lee et al., 2015). Given that the
monthly mean precipitation in May is only about 1 mm day$^{-1}$ (Fig. 3a), the reduction caused by
aerosols is about 20%. For monthly mean precipitation, changes over the mountain tops and the
lower elevation area, respectively, have similar seasonal variation patterns (Figs. 14b and 14c).

For SWE, however, changes over the mountain tops are different from those in the lower

area (Fig. 15). For mountain tops, negative changes in SWE are seen over the whole time period,
with a maximum reduction of about 60 mm in May corresponding to the maximum AOD (Fig.
15b). Major contribution is from local dust aerosols through ASI, as well as transported and local
anthropogenic aerosols through ACI. ARI produces small positive changes (~ 5 mm in May) in
SWE due to the scattering and absorption of solar radiation by aerosols which leads to surface
cooling. For lower elevation area, slightly enhanced SWE is found during the winter time,
associated with the effects of transported aerosols which produce more clouds through ACI, and
together with the ARI effect, lead to the cooling of the surface and hence less snowmelt (Fig. 15c).
Over the whole domain, SWE is reduced with a maximum of about 2 mm in May, equivalent to
about 2% reduction, mainly attributed to the local dust particles through ASI, and local
anthropogenic and transported aerosols through ACI (Fig. 15a).
Changes in temperature also exhibit different patterns over the mountain tops and the lower
elevations (Fig. 16). Warming over the mountain tops is produced by dust aerosols through ASI
with a maximum around May, and by transported aerosols through ACI during winter which leads
to reduced precipitation and SWE with a maximum in January (Fig. 16b). Cooling over the lower
elevation areas is caused by ARI, and also induced by more clouds generated in the model
simulations due to transported aerosols through ACI, with a maximum cooling of about 0.3 K in
April, corresponding to the maximum AOD of transported aerosols (Fig. 16c). The average
temperature changes over the whole domain are negative because of the large area of the lower
elevations (Fig. 16a).
Surface runoff reaches a maximum in December for the lower elevations and the whole
domain, but a peak value in May for mountain tops when the temperature is warmer (Fig. S4). For
lower elevations where there is not much snow, surface runoff is mainly associated with
precipitation and the changes present a similar pattern to those in precipitation (Fig. 17c). Changes
in surface runoff for the whole area present similar patterns to those of the lower elevations because
of the larger area of lower elevations (Fig. 17a). However for mountain tops, changes in surface
runoff are also associated with changes in snowmelt. Surface runoff over the mountain tops shows
a slight increase in spring, and then a decrease after April (Fig. 17b). The increase can be explained
by the effect of local dust aerosols deposited on the snow, which reduces the snow albedo through
ASI and warms the surface, leading to more and earlier snowmelt than normal, consistent with
negative changes in SWE. The decrease after April is a combined effect of less snowpack available
for melting caused by earlier snowmelt due to local dust aerosols and reduced precipitation caused
by transported and local anthropogenic aerosols through ACI. Thus, the impact of aerosols is to
speed up snowmelt at the mountain tops in spring and modify the seasonal cycle of surface runoff.

**4.    Conclusions**
A fully coupled high-resolution aerosol-meteorology-snowpack model is employed to
investigate the impacts of various aerosol sources on precipitation and snowpack in California.
The relative roles of locally emitted anthropogenic and dust aerosols, and aerosols transported
from outside the model domain are differentiated through the three pathways, aerosol-radiation
interaction (ARI), aerosol-snow interaction (ASI), and aerosol-cloud interaction (ACI). In the
following summary, the numbers in brackets represent the domain averaged changes (Table 4).
**Temperature**: Local dust aerosols warm the mountain top surface through ASI (0.12 K), in
which the reduced snow albedo associated with dirty snow leads to more surface absorption of
solar radiation. Transported and local anthropogenic aerosols warm the surface of mountain tops
through ACI (0.17 K), which produces more non-precipitating clouds but reduces precipitation
and hence snow amount, leading to decreased surface albedo and more absorption of solar energy.
The cooling of the valley area (−0.21 K) is primarily caused by the scattering and absorption of all
aerosols through ARI (−0.14 K). Transported and anthropogenic aerosols can also cool the surface
over the central valley through ACI (−0.07 K) that enhances cloud amount, leading to more
reflection of solar radiation.
**Precipitation and SWE**: Reduced precipitation of −6.87% is found due to the aerosol effects
and is mainly caused by transported and local anthropogenic aerosols through ACI (−6.26%). The
maximum of aerosol effect on precipitation is found in December during the rainy season when
the aerosols loadings are also relatively large. The other peak effect occurs in May with a reduction
of about 20%, probably associated with the maximum of aerosol loadings and more orographic
precipitation over the mountains. Locally emitted dust aerosols represent one of the most important
contributors to the reduced SWE (−3.17%) through ASI (−1.96%), with the largest reduction in
May corresponding to the maximum dust emission over that time. Local anthropogenic aerosols
can also reduce SWE through ACI (−2.67%). On the other hand, ARI (2.43%) by all aerosols, with
most contributions from the transported aerosols, exceeds the effects of ASI (−0.99%) and ACI
(−0.27%) and slightly enhance SWE by 0.42% over lower elevations in winter time through the
surface cooling.
**Surface runoff**: As a major component of the water cycle, surface runoff is mainly generated
by precipitation, but for mountain tops, the changes in surface runoff are also associated with the
changes in snowmelt. We find that the seasonal-mean surface runoff is reduced by −6.58%
associated with suppressed precipitation, caused by transported and anthropogenic aerosols
through ACI (−6.30%). Over mountain tops, runoff slightly increases in spring due to the enhanced
solar absorption by dust aerosols. Runoff decreases after April as a combined effect of less
snowpack available for melting caused by earlier snowmelt due to local dust and reduced
precipitation due to transported and local anthropogenic aerosols through ACI. Therefore, one of
the important impacts of aerosols is to speed up the snowmelt at mountain tops in spring and
modify the seasonal cycle of surface runoff.
In summary, we find that the WRF-Chem model simulations with aerosol effects included
would produce lower precipitation and SWE by about 10% and colder temperature by 0.2 K over
California than the simulations without aerosols. Therefore, including aerosol effects can reduce
the high biases of these variables in the simulations reported previously. Aerosols play an
important role in California water resources through the warming of mountain tops and the
subsequent modification of precipitation and snowmelt. The total aerosol effects produce a
warming of 0.22 K over mountain tops and a reduction from October to June in precipitation, SWE,
and surface runoff of about 7%, 3%, and 7%, respectively, for the whole domain, with
corresponding numbers of 10% or more over mountain tops. In a dry year (year 2013-2014),
aerosol can have more significant impacts on SWE, with a reduction of up to 9% for the whole
domain and 16% over mountain tops.

It is challenging to accurately represent aerosol properties in the model (Fast et al., 2014).

As pointed out by Wu et al. (2017), biases exist in the current model as compared to observations,
for example, underestimation of AOD due to poor representation of dust emission and vertical
mixing in the warm season. The underestimate of AOD in the model implies that the simulated
aerosol effects could also be biased low. Given the important role that dust plays in the California
snowpack, improved dust emission and vertical mixing are needed for accurate quantification of
the impact of dust. Also, the underestimation of organic matter (associated with secondary organic
aerosol processes) in the model (Wu et al., 2017), which are primarily scattering aerosols, would
contribute to the high bias in the simulation of surface temperature. More accurate representation
and simulation of these aerosols in the model are needed. In the current WRF-Chem model, the
aerosol effect on ice clouds is not included. ACI associated with ice clouds are more complex than
that with liquid clouds. For example, a few studies have shown that negative Twomey effects may
occur with aerosols and ice clouds, in which increased aerosols (and thus ice nuclei) lead to
enhanced heterogeneous nucleation that is associated with larger and fewer ice crystals as
compared to the homogeneous nucleation counterpart (DeMott et al., 2010; Chylek et al., 2006,
Zhao et al. 2018). A recent study shows that the responses of ice crystal effective radius to aerosol
loadings are modulated by water vapor amount in conjunction with several other meteorological
parameters. While there is a significant negative correlation between ice effective radius and
aerosol loading in moist conditions, consistent with the "Twomey effect" for liquid clouds, a strong
positive correlation between the two occurs in dry conditions (Zhao et al. 2018). Despite numerous
studies about the impact of aerosols on ice clouds, the role of anthropogenic aerosols in ice
processes, especially over polluted regions, remains a challenging scientific issue. The effect of
anthropogenic aerosols on ice formation and cloud radiative properties may be a critical pathway
through which anthropogenic activities affect regional climate and present the opportunities for
further studies using observations and models.

Our model simulation produces relative larger SWE than the SNOTEL observations.

Improvement of snowpack simulation in the land surface model is needed for accurate
quantification of aerosol impacts on snowpack. Our results are based on two years of simulations.
Additional simulations under different meteorological conditions will help to better assess the
aerosol impacts on California hydrology quantitatively.

**Data availability**
The PRISM data are available through the following link: http://prism.oregonstate.edu/recent/.
The     CPC     data     are     available     through     the     following     link:
https://www.esrl.noaa.gov/psd/data/gridded/data.unified.daily.conus.html. The DWR data are
available through the following link: http://cdec.water.ca.gov/snow_rain.html. The CIMIS data are
available through the following link: http://wwwcimis.water.ca.gov/. The SNOTEL data are
available through the following link: https://www.wcc.nrcs.usda.gov/snow/. The MISR data is

available through the following link: https://misr.jpl.nasa.gov/getData/accessData/. The NLDAS

MOS0125 albedo data are available through the following link:

https://giovanni.gsfc.nasa.gov/giovanni/#service=TmAvMp&starttime=&endtime=&variableFac

ets=dataFieldMeasurement%3AAlbedo%3BdataProductPlatformInstrument%3ANLDAS%20M

odel%3BdataProductTimeInterval%3Amonthly%3B.

## Competing interests

The authors declare that they have no conflict of interest.

## Acknowledgements

This study was carried out at the Joint Institute for Regional Earth System Science and Engineering and Department of Atmospheric and Oceanic Science, University of California, Los Angeles, and sponsored by California Energy Commission under grant #EPC-14-064. LW, JHJ, HS, and YSC conducted the work at the Jet Propulsion Laboratory, California Institute of Technology, under contract with the National Aeronautics and Space Administration. They acknowledge the funding support from the NASA ACMAP program. CZ is supported by the "Thousand Talents Plan for Young Professionals" program of China. The contribution of YQ is supported by the U.S. Department of Energy (DOE), Office of Science, Biological and Environmental Research as part of the Regional and Global Climate Modeling Program. The Pacific Northwest National Laboratory (PNNL) is operated for DOE by Battelle Memorial Institute under contract DE-AC05-76RL01830. We would like to thank two anonymous reviewers for their constructive comments and suggestions for the improvement of this paper.

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

Table 1. Model configuration

| Atmospheric Process | WRF-Chem option |
|---|---|
| Microphysics | Morrison double-moment |
| Radiation | RRTMG for both shortwave and longwave |
| Land surface | CLM4 with SNICAR included |
| Planetary boundary layer (PBL) | YSU |
| Cumulus | No cumulus scheme used |
| Chemical driver | CBM-Z |
| Aerosol driver | MOSAIC 4-bin |
| Anthropogenic emission | NEI05 |
| Biogenic emission | MEGAN |
| Biomass burning emission | GFEDV2.1 |
| Dust emission | DUSTRAN |
| Meteorological initial and boundary conditions | ERA-Interim |
| Chemical initial and boundary conditions | MOZART-4 divided by 2 |


Table 2. Experiment design for various aerosol sources.

| Experiment | Anthropogenic Aerosols | Dust Aerosol | Transport | Description |
|---|---|---|---|---|
| CTRL | Y | Y | Y | Control experiment with all aerosol emissions/transports included |
| NoLocDust | Y | N | Y | Local dust aerosol emission is not included |
| NoLocAnth | N | Y | Y | Local anthropogenic aerosol emissions are not included |
| NoTran | Y | Y | N | Aerosols transported from outside the model domain are not included |
| CLEAN | N | N | N | Aerosol emissions/transports are not included |


Table 3. Experiment design for various aerosol pathways, using the CTRL aerosol emissions.

| Experiment | ARI | ACI | ASI | Description |
|---|---|---|---|---|
| NARI | N | Y | Y | ARI is not included |
| NASI | Y | Y | N | ASI is not included |
| NARS | N | Y | N | ARI and ASI are not included |


Table 4. Changes in surface temperature (K) and precipitation, SWE, and surface runoff in
percentage averaged over October 2012 to June 2013 due to overall and various aerosol effects for
the whole domain (34-42 °N, 117-124 °W, not including ocean points), mountain tops (with
elevation ≥ 2.5 km), and lower elevations ( < 2.5 km). Total impacts for the simulations from
October 2013 to June 2014 are also included as "Total_13-14".

| Region | Source/ pathway | T2 (K) | Precipitation (%) | SWE (%) | Surface runoff (%) |
|---|---|---|---|---|---|
| Whole Domain | Total | −0.19 | −6.87 | −3.17 | −6.58 |
| | Total_13-14 | −0.21 | −5.99 | −8.88 | −5.13 |
| | ARI | −0.14 | −0.47 | 1.88 | −0.21 |
| | ASI | 0.01 | −0.03 | −1.96 | 0.04 |
| | ACI | −0.06 | −6.26 | −2.67 | −6.30 |
| | LocAnth | −0.02 | −1.02 | −0.91 | −0.94 |
| | LocDust | 0.00 | −0.19 | −1.35 | 0.01 |
| | Tran | −0.17 | −2.97 | 1.89 | −2.90 |
| Mountain Tops | Total | 0.22 | −11.53 | −10.50 | −9.58 |
| | Total_13-14 | 0.15 | −9.90 | −15.57 | −3.55 |
| | ARI | −0.09 | −0.61 | 0.76 | −0.49 |
| | ASI | 0.12 | 0.26 | −3.94 | 1.10 |
| | ACI | 0.17 | −11.03 | −7.57 | −10.25 |
| | LocAnth | 0.03 | −1.75 | −1.60 | −2.06 |
| | LocDust | 0.10 | 0.31 | −2.99 | 1.49 |
| | Tran | −0.02 | −5.25 | −2.43 | −4.76 |
| Lower Elevations | Total | −0.21 | −6.62 | 0.42 | −6.42 |
| | Total_13-14 | −0.22 | −5.75 | 2.66 | −5.26 |
| | ARI | −0.14 | −0.46 | 2.43 | −0.19 |
| | ASI | 0.00 | −0.04 | −0.99 | −0.01 |
| | ACI | −0.07 | −6.00 | −0.27 | −6.09 |
| | LocAnth | −0.03 | −0.98 | −0.57 | −0.89 |
| | LocDust | 0.00 | −0.22 | −0.55 | −0.07 |
| | Tran | −0.17 | −2.85 | 4.01 | −2.81 |


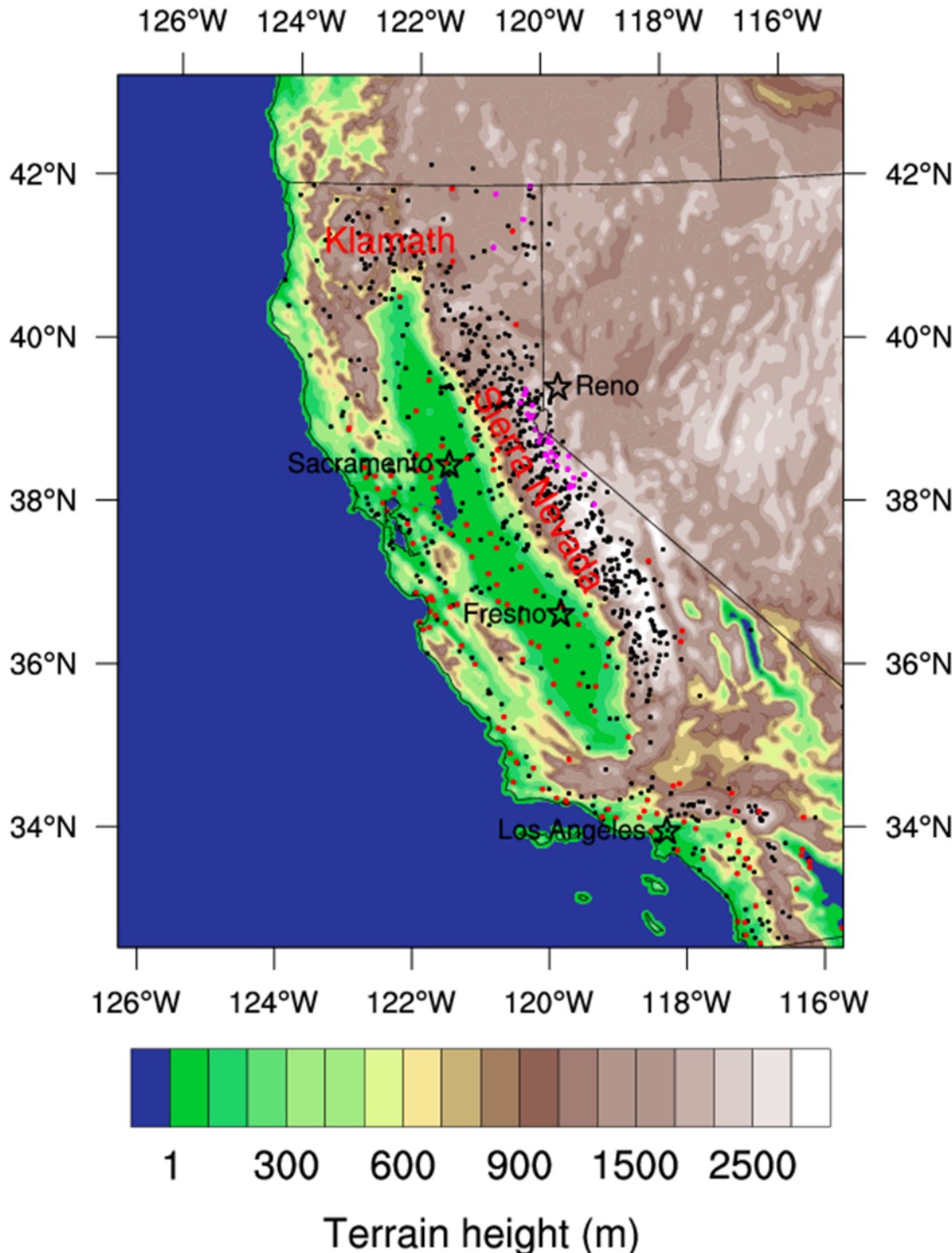

Figure 1. Model domain and terrain height (m). 991 DWR sites are represented by black dots; 138 CIMIS stations are represented by red dots; 32 SNOTEL sites are represented by magenta dots.

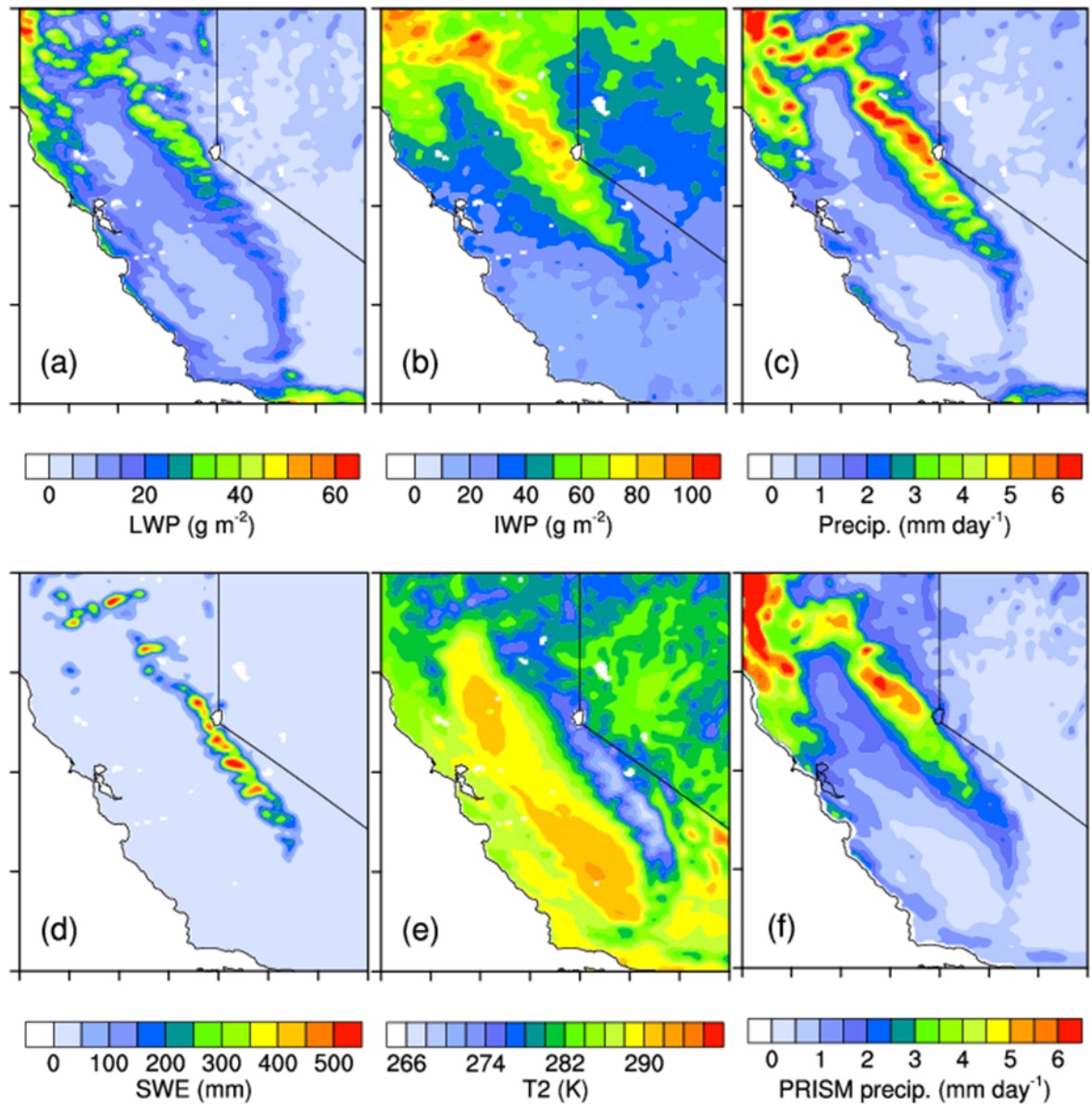

925

Figure 2. Model simulated (a) LWP (g m$^{-2}$), (b) IWP (g m$^{-2}$), (c) precipitation (mm day$^{-1}$), (d) SWE

(mm), and (e) temperature at 2 meters, T2 (K) from the CTRL simulation, and (f) PRISM observed

precipitation (mm day$^{-1}$), averaged over October 2012 to June 2013.

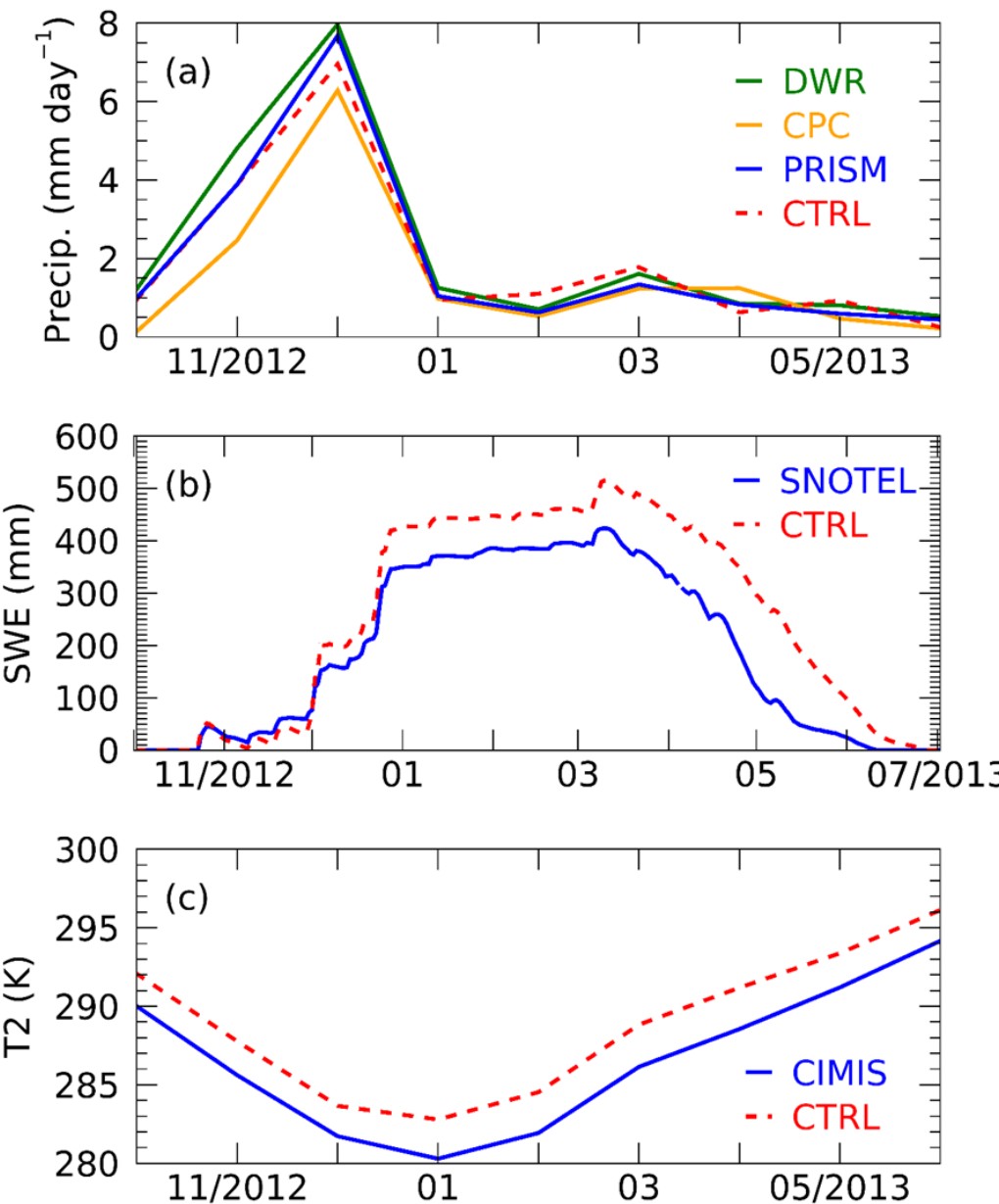

929

Figure 3. (a) Monthly mean precipitation (mm day$^{-1}$) from the CTRL simulation (red dashed) and

PRISM (blue), CPC (orange) and DWR (green) observations; (b) Daily accumulated SWE (mm)

from the CTRL simulation (red dashed) and SNOTEL observation (blue); and (c) Monthly mean

T2 (K) from the CTRL simulation (red) and CIMIS observation (blue). Model data are sampled

onto observational sites before the comparison is conducted.

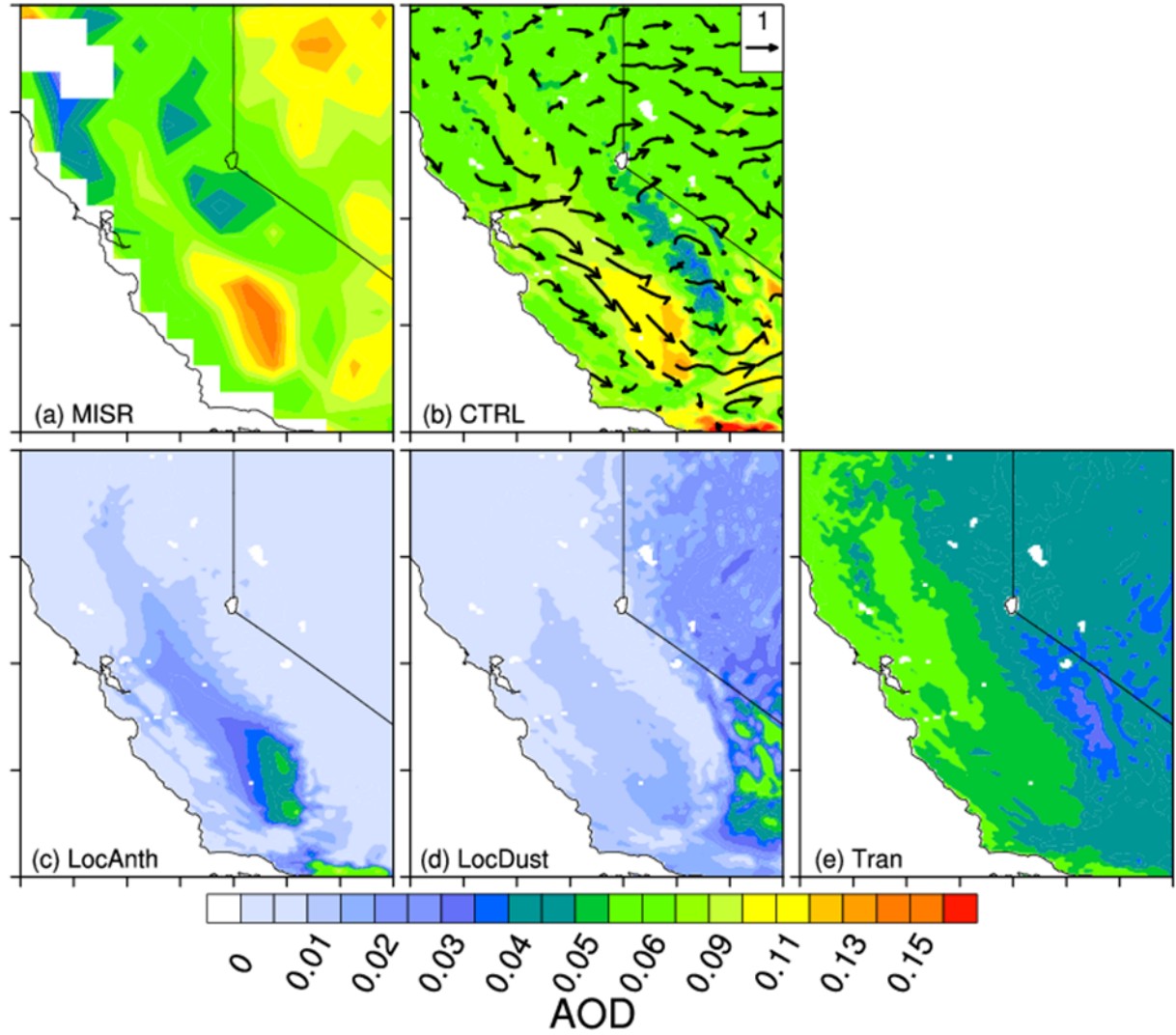

935

Figure 4. Spatial distribution of aerosol optical depth (AOD) averaged over October 2012 to June

2013 for (a) MISR observations, (b) all aerosols in the CTRL simulation, (c) local anthropogenic

aerosols, (d) local dust aerosols, and (e) transported aerosols from outside the domain, derived

from the difference between the CTRL simulation and the corresponding experiment (NoLocAnth,

NoLocDust and NoTran), respectively. 10-m wind vectors from the CTRL simulation is shown in

(b).

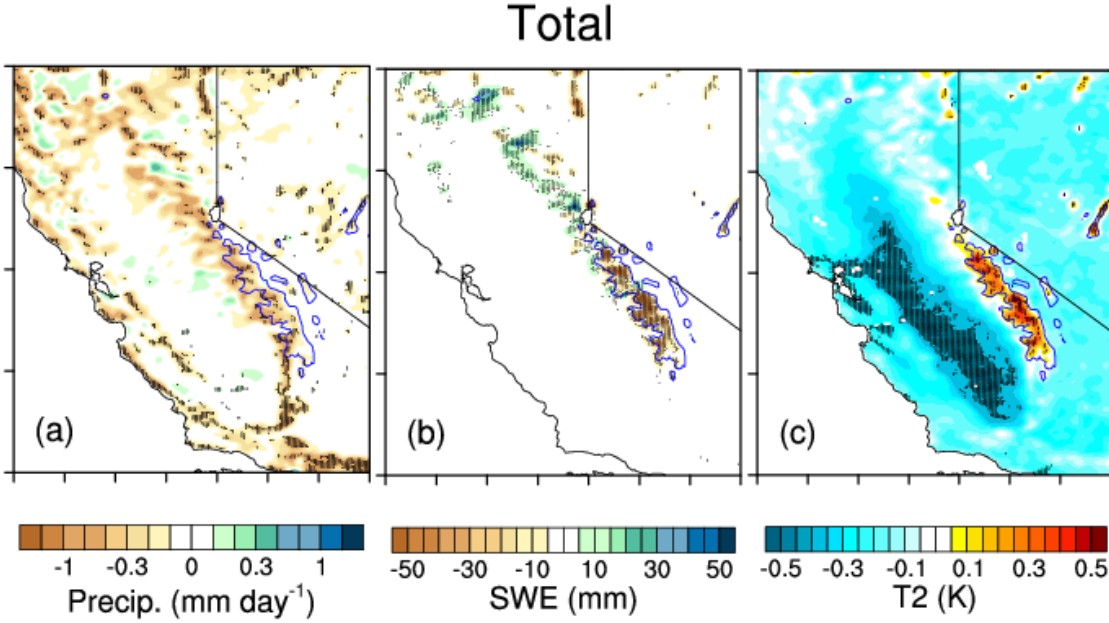


Figure 5. Total aerosol effects (CTRL – CLEAN) on spatial distribution of (a) precipitation (mm

day⁻¹), (b) SWE (mm), and (c) T2 (K). The dotted area denotes statistical significance above the

90% confidence level. Blue lines represent the mountain tops with elevation ≥ 2.5 km.

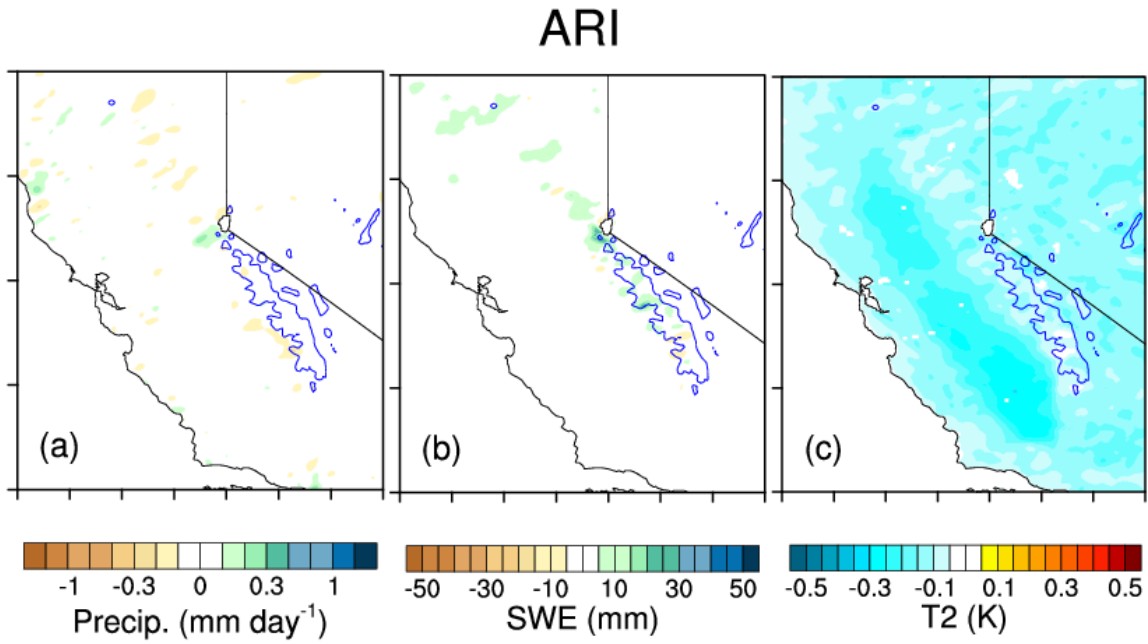


Figure 6. ARI effects (CTRL – NARI) on spatial distribution of (a) precipitation (mm day$^{-1}$), (b)
SWE (mm), and (c) T2 (K). Blue lines represent the mountain tops with elevation $\geq$ 2.5 km.

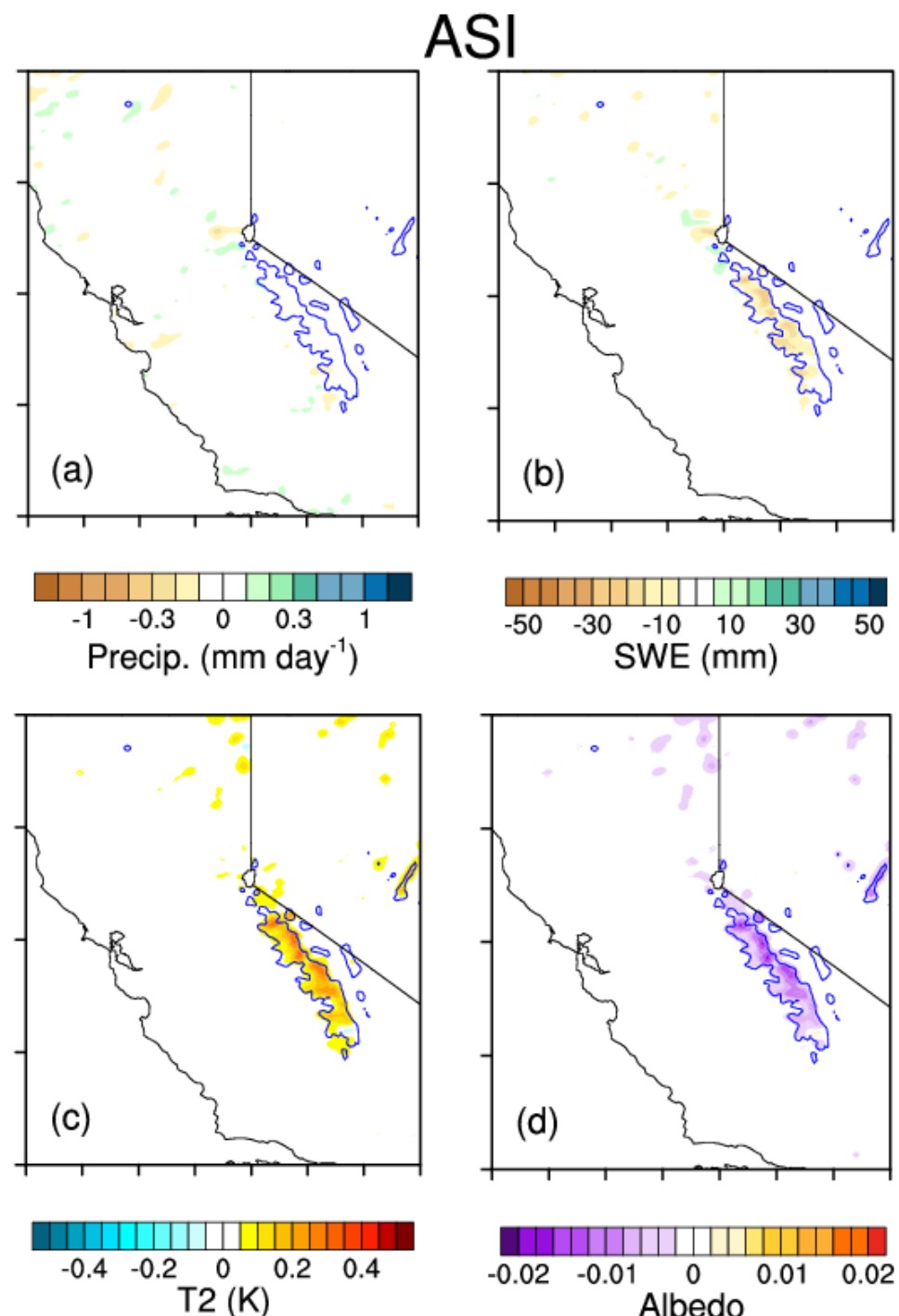


Figure 7. ASI effects (CTRL – NASI) on spatial distribution of (a) precipitation (mm day[-1]), (b)
SWE (mm), (c) T2 (K), and (d) surface albedo. Blue lines represent the mountain tops with
elevation $\geq$ 2.5 km.

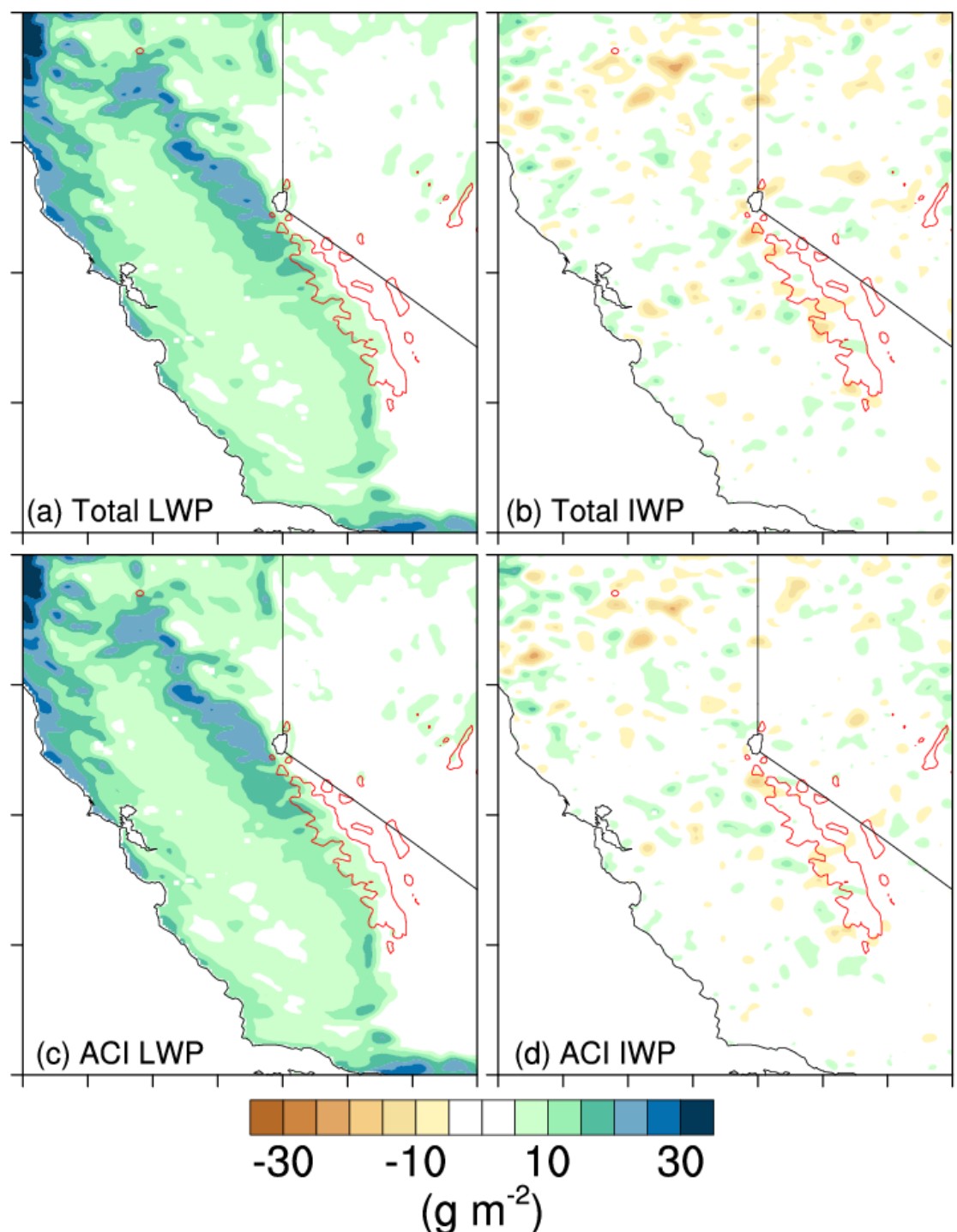


Figure 8. Differences in (a) LWP (g m$^{-2}$) and (b) IWP (g m$^{-2}$) due to all aerosol effects (CTRL –
CLEAN), and (c) LWP (g m$^{-2}$) and (d) IWP (g m$^{-2}$) due to ACI effect (NARS – CLEAN). Red
lines represent the mountain tops with elevation $\geq$ 2.5 km.

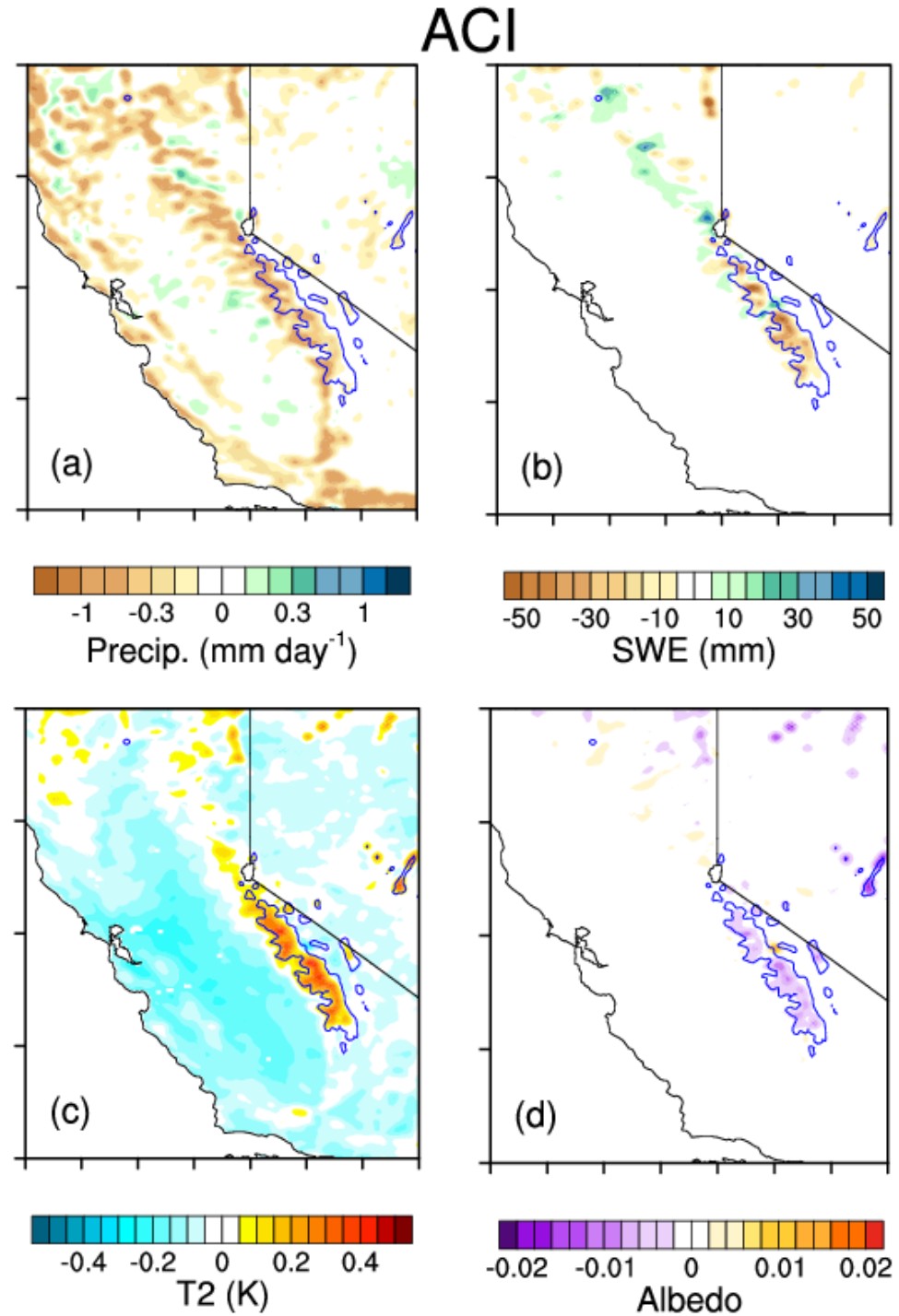


Figure 9. Same as Figure 7, but for ACI effect (NARS – CLEAN).

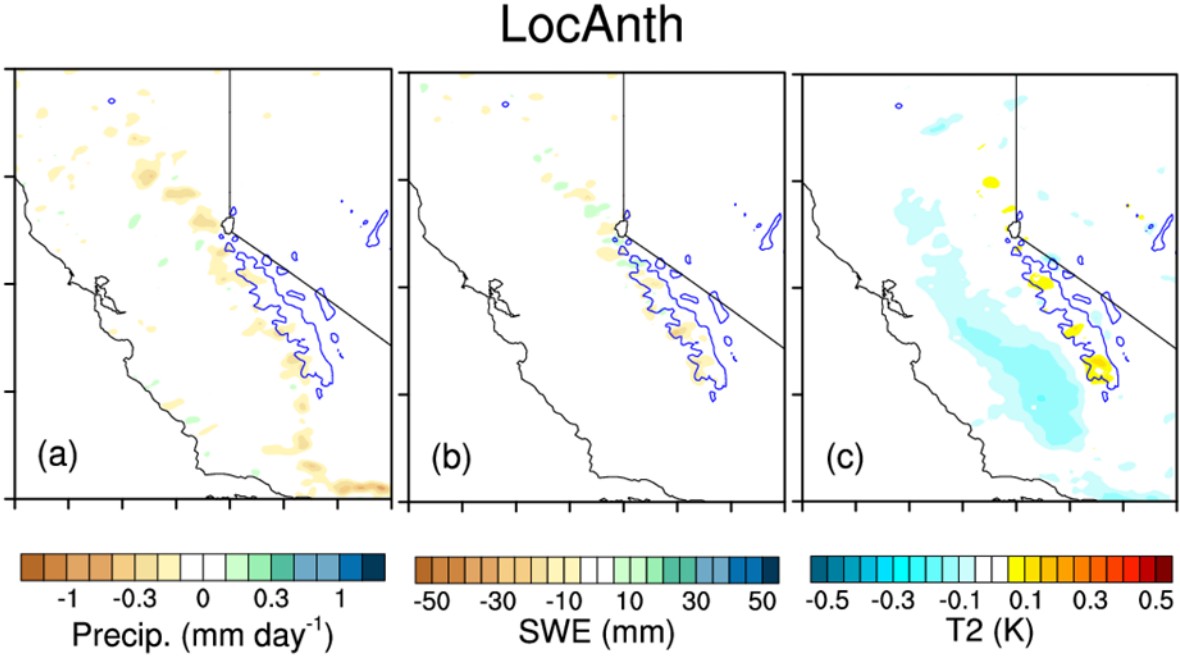


Figure 10. Effect of local anthropogenic aerosols (CTRL – NoLocAnth) on spatial distribution of
(a) precipitation (mm day$^{-1}$), (b) SWE (mm), and (c) T2 (K). Blue lines represent the mountain
tops with elevation ≥ 2.5 km.

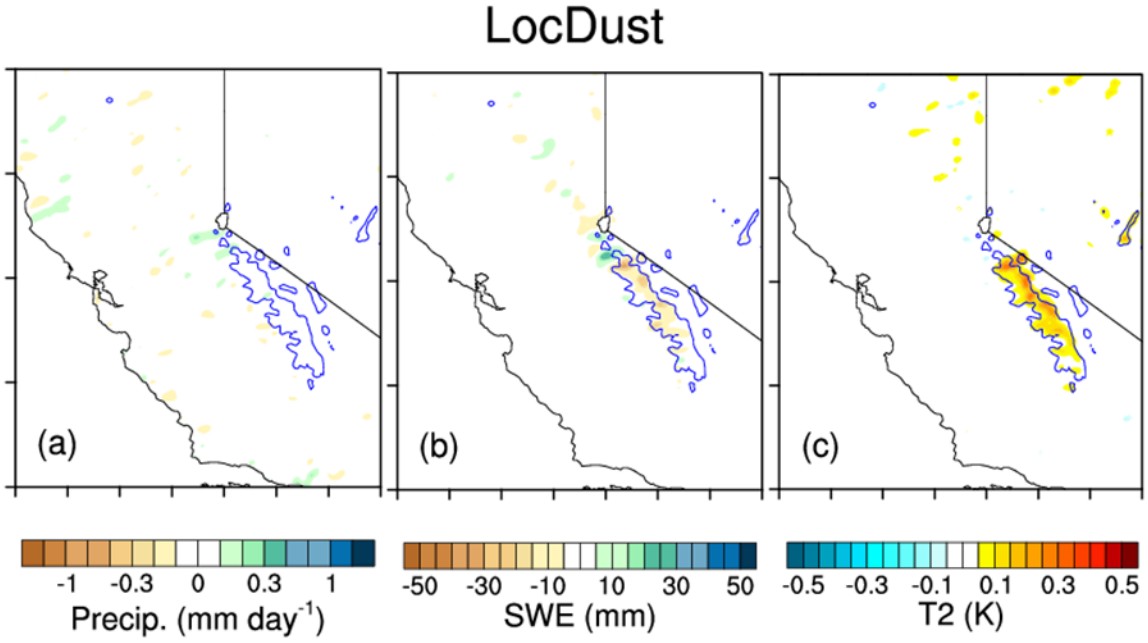

LocDust

Precip. (mm day$^{-1}$)

SWE (mm)

T2 (K)


Figure 11. Same as Figure 10, but for the effect of local dust aerosols (CTRL – NoLocDust).

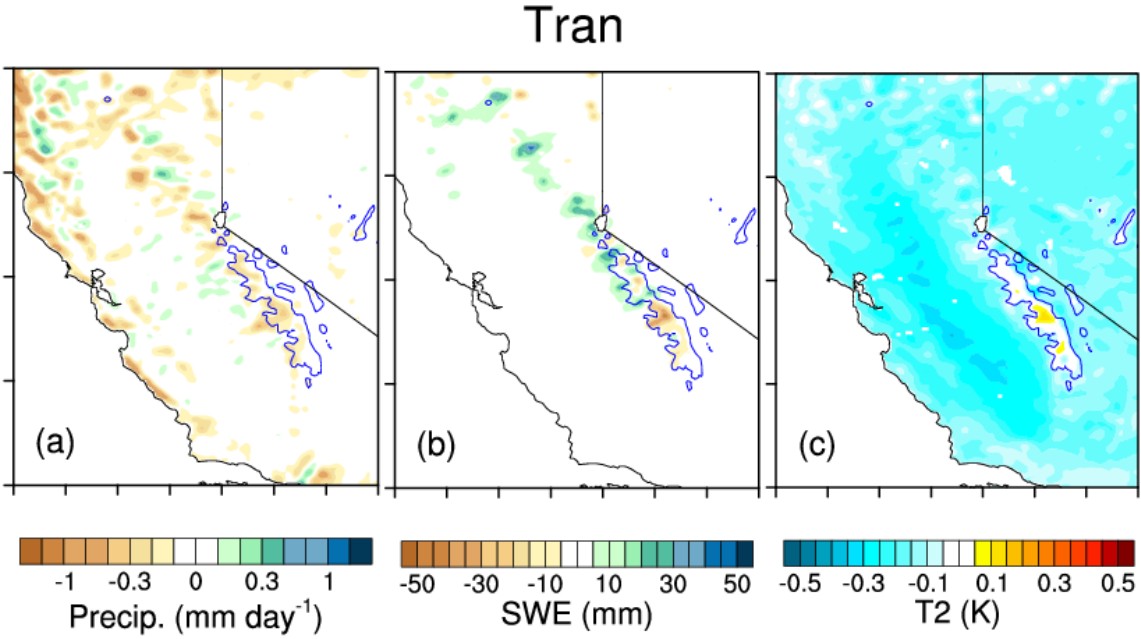


Figure 12. Same as Figure 10, but for the effect of transported aerosols (CTRL – NoTran).

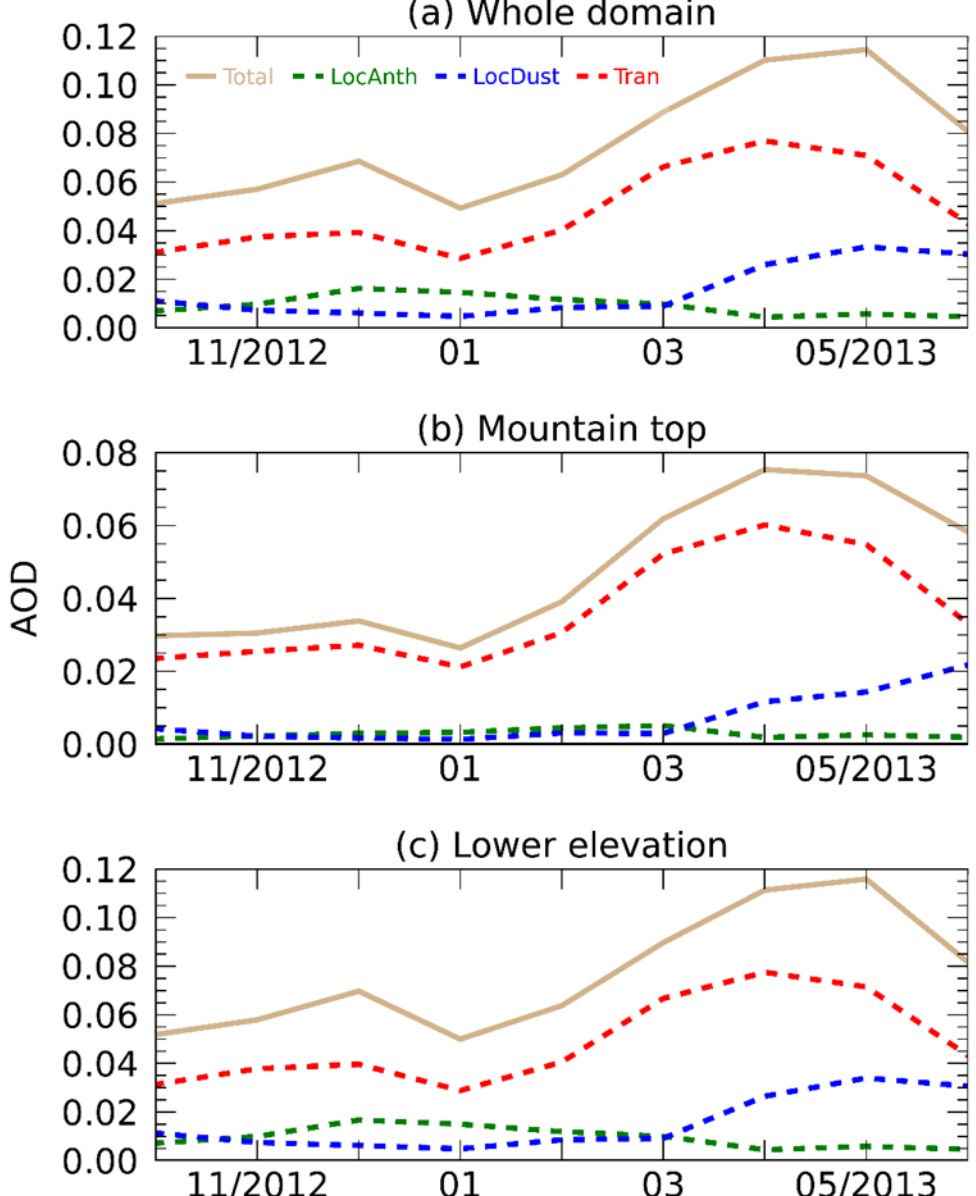


Figure 13. Monthly mean AOD simulated from CTRL for total aerosols (brown solid), local anthropocentric aerosols (green dashed), local dust (blue dashed), and transported aerosols (red dashed) averaged over (a) the whole domain (34-42 °N, 117-124 °W, not including ocean points), (b) mountain tops (with elevation $\geq 2.5$ km), and (c) lower elevation area ($< 2.5$ km) from October 2012 to June 2013.

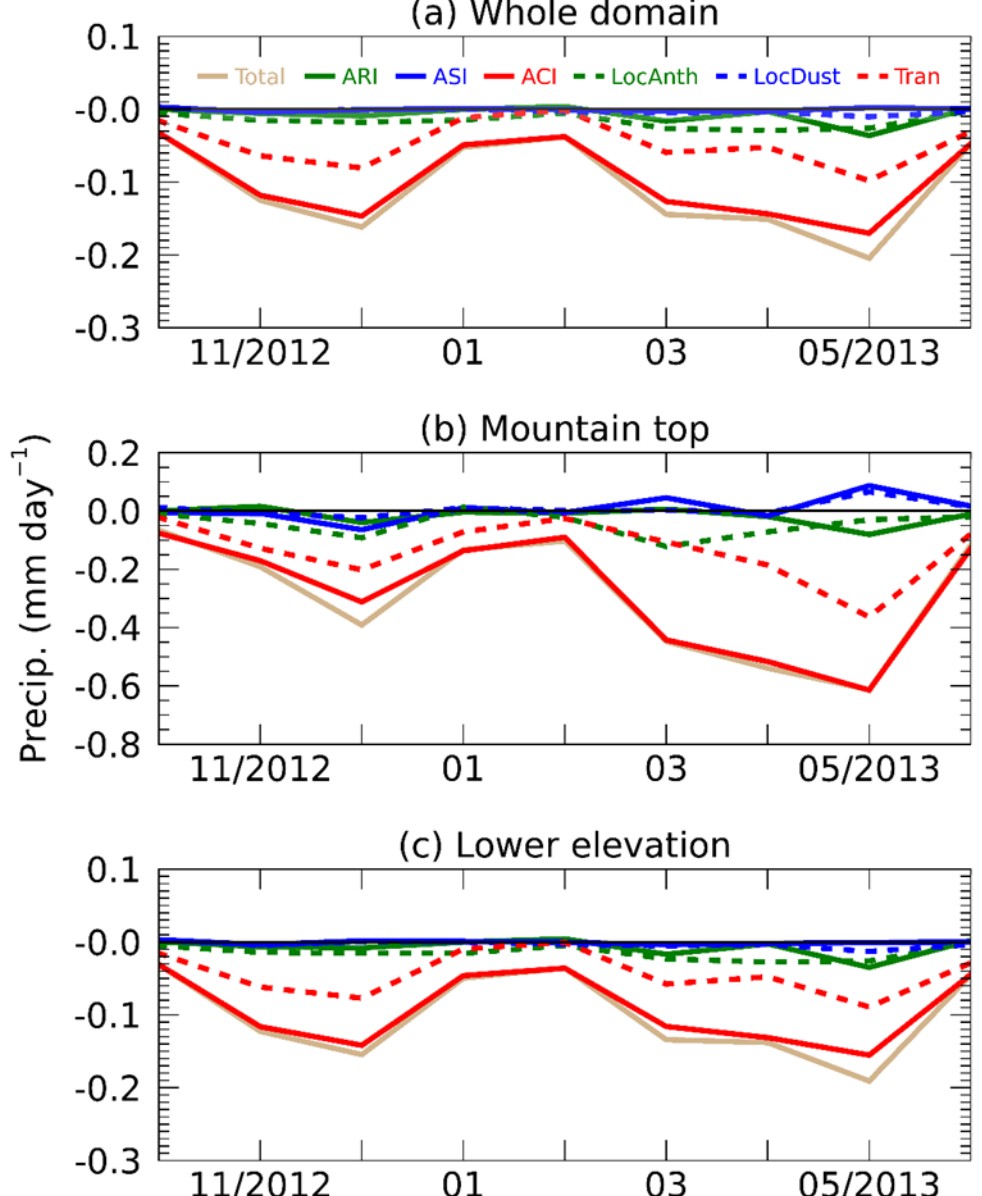


Figure 14. Monthly mean differences in precipitation (mm day$^{-1}$) due to total aerosols (brown
solid), ARI (green solid), ASI (blue solid), ACI (red solid), local anthropocentric aerosols (green
dashed), local dust (blue dashed), and transported aerosols (red dashed) averaged over (a) the
whole domain (34-42 °N, 117-124 °W, not including ocean points), (b) mountain tops (with
elevation ≥ 2.5 km), and (c) lower elevation area ( < 2.5 km) from October 2012 to June 2013.
Zero line is shown as thin black line.

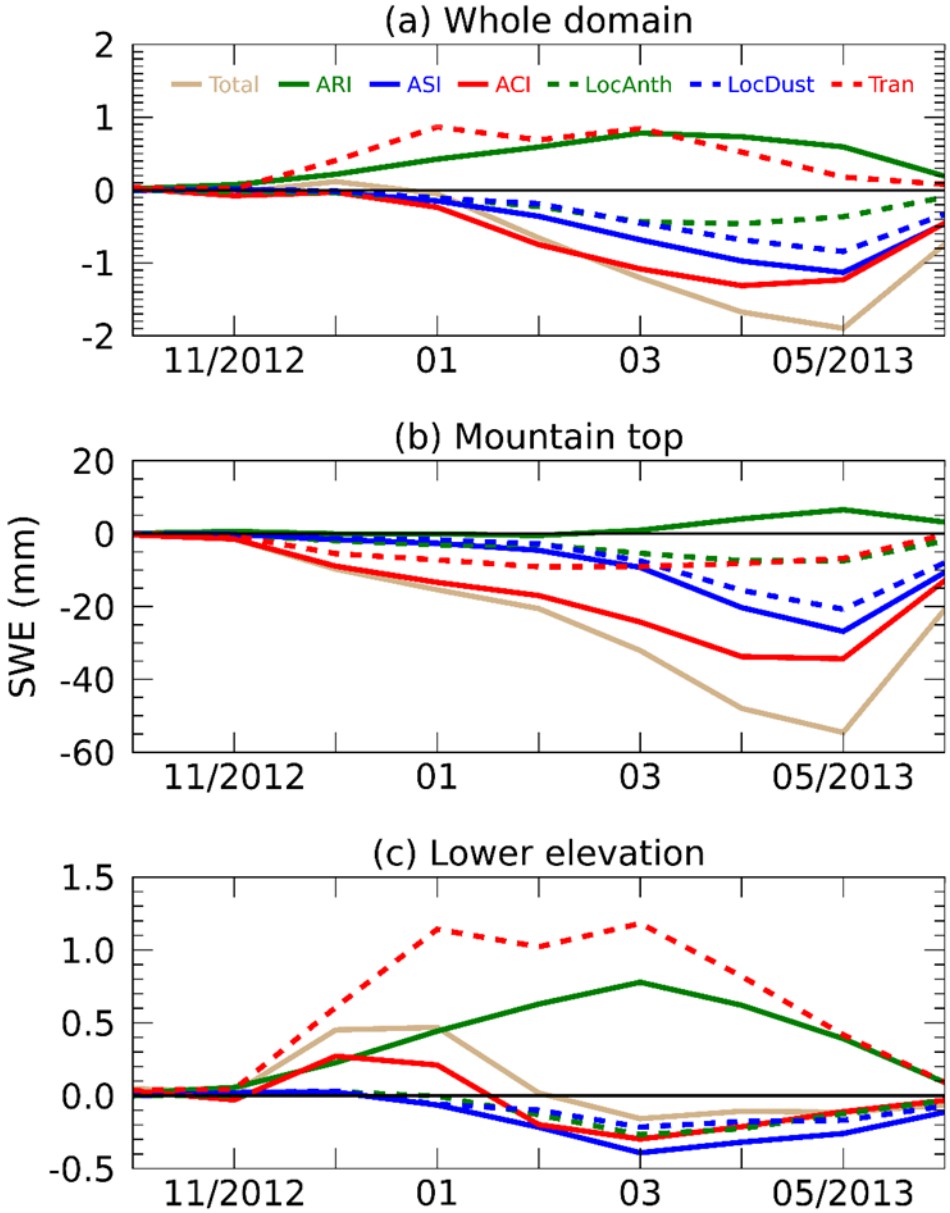


Figure 15. Same as Figure 14, but for SWE (mm).

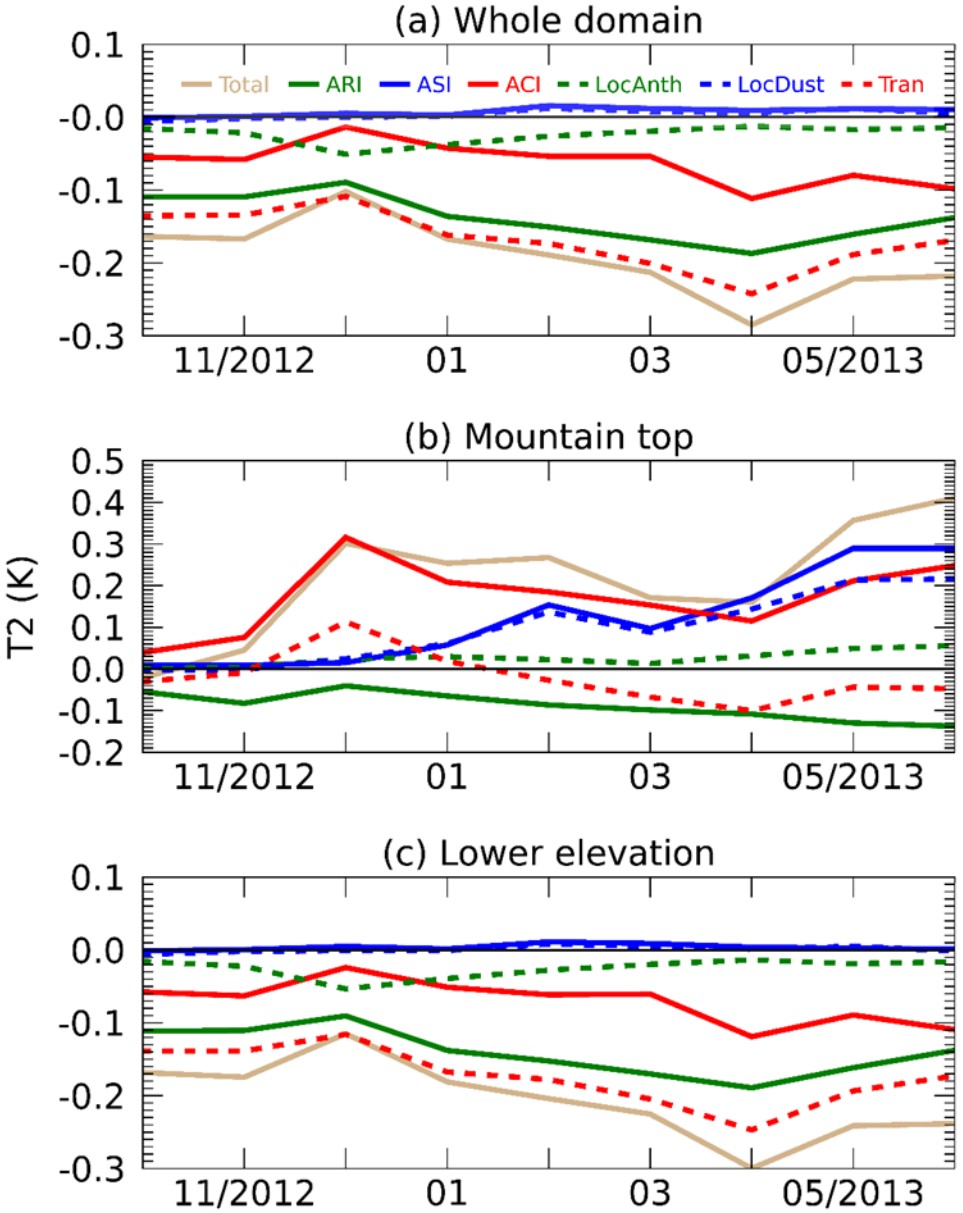


Figure 16. Same as Figure 14, but for T2 (K).

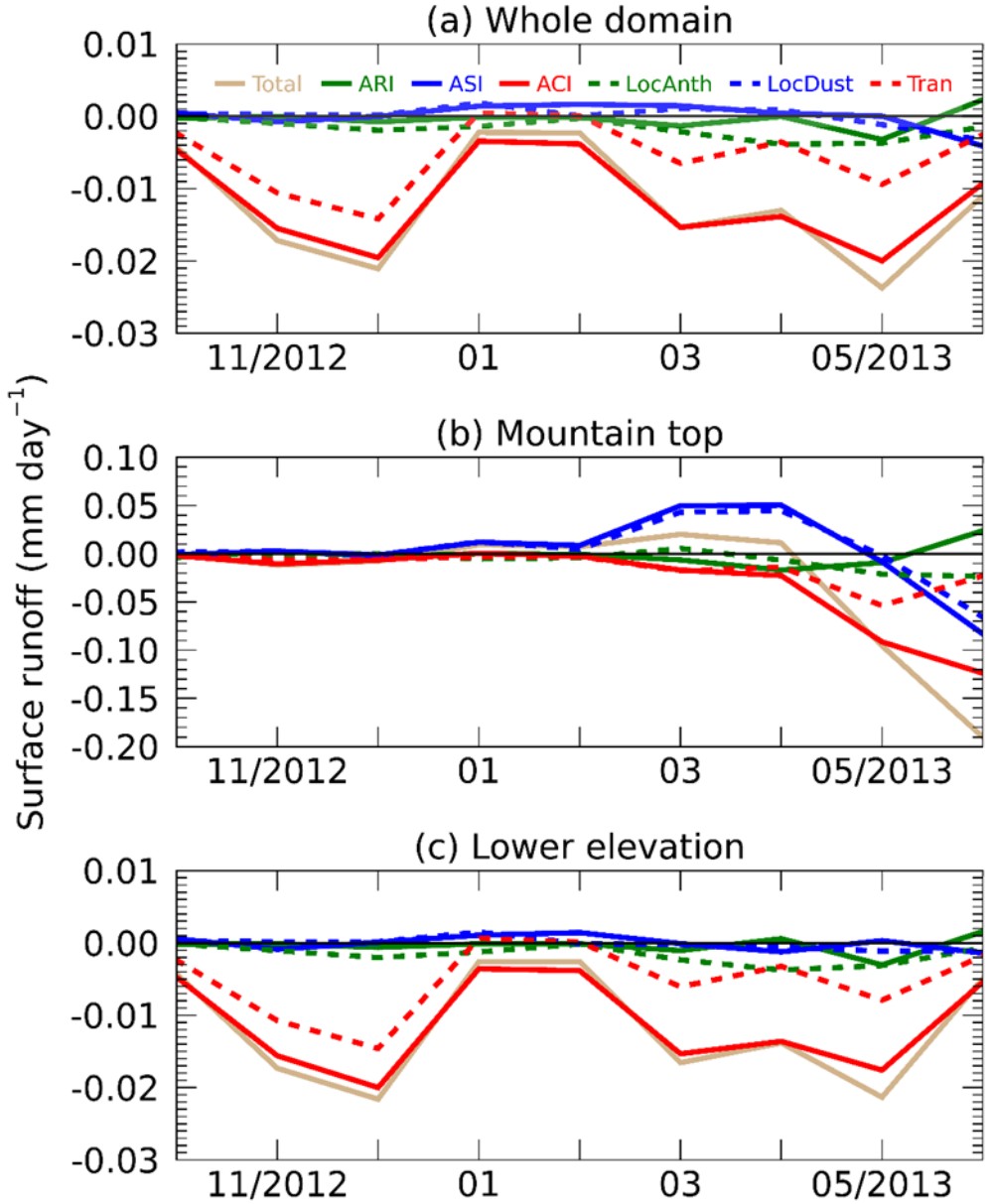


Figure 17. Same as Figure 14, but for surface runoff (mm day$^{-1}$).