# Peer review of "Impacts of Aerosols on Seasonal Precipitation and Snowpack in California 1 Based on Convection-Permitting WRF-Chem Simulations 2 3 Longtao Wu1, Yu Gu2, Jonathan H. Jiang1, Hui Su1, Nanpeng Yu3, Chun Zhao4, Yun Qian5, Bin 4 Zhao2, Kuo-Nan Liou2, and Yong-Sang Choi1,6 5 6 1Jet Propulsion Laboratory, California Institute of Technology, Pasadena, CA, USA. 7 2Joint Institute for Regional Earth System S"

_Atmospheric Chemistry and Physics, 2017_

## Short Comment (SC1) · 25 Oct 2017

Can the authors motivate dividing the MOZART-4 chemical boundary conditions by a factor of 2 for this study (Table 1)?

---

## Author Comment (AC1) · 25 Oct 2017

Fast et al. (2014) and Wu et al. (2017) found that the MOZART-4 model overestimates aerosols in the free troposphere over California. With MOZART-4 boundary conditions divided by 2, aerosols in the WRF-Chem simulations have better agreement with observations in California.

Fast, J. D., Allan, J., Bahreini, R., Craven, J., Emmons, L., Ferrare, R., Hayes, P. L., Hodzic, A., Holloway, J., Hostetler, C., Jimenez, J. L., Jonsson, H., Liu, S., Liu, Y., Metcalf, A., Middlebrook, A., Nowak, J., Pekour, M., Perring, A., Russell, L., Sedlacek, A., Seinfeld, J., Setyan, A., Shilling, J., Shrivastava, M., Springston, S., Song, C., Subra-

manian, R., Taylor, J. W., Vinoj, V., Yang, Q., Zaveri, R. A., and Zhang, Q.: Modeling regional aerosol and aerosol precursor variability over California and its sensitivity to emissions and long-range transport during the 2010 CalNex and CARES campaigns, Atmos. Chem. Phys., 14, 10013-10060, https://doi.org/10.5194/acp-14-10013-2014, 2014.

Wu, L., Su, H., Kalashnikova, O. V., Jiang, J. H., Zhao, C., Garay, M. J., Campbell, J. R., and Yu, N.: WRF-Chem simulation of aerosol seasonal variability in the San Joaquin Valley, Atmos. Chem. Phys., 17, 7291-7309, https://doi.org/10.5194/acp-17-7291-2017, 2017.

---

## Referee Comment (RC1) · Anonymous Referee #1 · 20 Nov 2017

Aerosols can induce large impacts on the regional climate and hydrologic cycles. Currently the aerosol effects are still not well understood, especially for the individual and combined effects of different underlying mechanisms (direct, indirect, and feedback).

This study presents a comparison of different aerosol effects including aerosol-radiation interaction (ARI), aerosol-cloud interaction (ACI), and aerosol-snow interaction (ASI) on the regional climate in California based on WRF-Chem simulations. The study also shows the different effects induced by local dust emissions, local anthropogenic emissions, and transportation. Overall, the manuscript is well written, and most of the content is well organized. The scientific findings are significant to our un-

derstanding of climatic effects of different aerosols. This study is useful for the relevant research community on unraveling the aerosol affects in climate and hydrologic cycles.

However, some statements are not clear and some of them may need further evidence. Part of the manuscript can be better organized for easy following. I have some suggestions and comments that I would like the authors to consider before the manuscript can be accepted for publication in ACP.

Major comments:

(1) Lines 254-256, Figure 3, Lines 36-40 (Abstract): The authors states that the model simulations represent reasonable magnitude of SWE, because SNOTEL data underestimates real SWE. They deduce the underestimate of SNOTEL SWE from "The main issue with weighing-type gauges for snowfall estimation is the undercatch of approximately 10%–15% due to wind (Serreze et al., 2001; Yang et al., 1998; Rasmussen et al., 2001). " (Lines 249-251). I should mention that snowfall is not SWE. They are measured differently: snowfall referring to a solid form of precipitation is measured by gauges, while SWE is measured using a snow pillow (https://www.wcc.nrcs.usda.gov/about/mon_automate.html). Therefore, underestimation of snowfall doesn't mean underestimation of SWE. If SNOTEL SWE is not underestimated compared to the reality, the model (with aerosol effects) may have large biases in SWE (up to ∼100 mm) (Figure 3b).

The authors state that inclusion of aerosol effects reduce the model biases (Abstract). Although it is generally true, it is not simply the case for a model simulation regarding the large uncertainties in current models. With aerosol effects, WRF-Chem reduces SWE biases by 0-60 mm (Figure 15), but still has the bias of ∼100 mm (mentioned above, if SNOTEL SWE is not biased low). Although the authors can still get the conclusion of reduction of SWE biases with aerosol effects, discussion on other reasons for the model biases (potentially larger than the biases that can be reduced by including aerosol effects) is desirable and helpful. In addition, model simulations are not

always improved with the inclusion of aerosols effects. For example, CTRL simulation underestimates precipitation in April (Figure 3a). If the aerosol effects are removed, simulated precipitation is larger (Figure 14), which is more consistent with the observation.

For precipitation and temperature, there are multiple observations available for comparison with the model simulations. Without the investigations of the reliability of each observation, the selected observations may be arbitrary. Besides the CPC, DWR, and CIMIS observations used in this study, there are also other datasets (including a widely-used dataset, PRISM-Parameter-elevation Regressions on Independent Slopes Model) available but not included. The resolution of PRISM (4 km), much higher than CPC (0.25 degree) used, is also similar to the model resolution (4km). I am wondering how the simulation results are compared to the PRISM observation at similar resolution.

Overall, more investigation is needed to support the improvement of model performance when aerosol effects are included, by comparison of model results with more observation datasets and consideration of the reliability of these observations.

(2) Table 3, Lines 216-223: The authors decompose the effects of ARI, ACI, and ASI from these multiple experiment. Do they assume the linear combination of ARI, ACI, and ASI? It is possible that ARI, ACI, and ASI can be interacted to generate overall effects. CTRL-NARI (CTRL-NASI) may include the interaction of ARI/ACI and ARI/ASI (ARI/ASI and ACI/ASI), which may be different from NASI-NARS (NARI-NARS). If any difference between CTRL-NARI and NASI-NARS (CTRL-NASI and NARI-NARS) is found, it is also helpful if the authors can explicitly mention this nonlinear combination of ARI/ACI/ASI. Although it is difficult to identify the interaction of ARI, ACI, and ASI, at least some discussions are needed.

(3) Section 2: The authors describe the three pathways of aerosol effects in the order of ARI, ACI, and ASI in Introduction, but describe their representation in WRF-Chem in the

order of ASI, ARI, and ACI in Section 2. This tends to give the readers an impression that ASI is more important than ARI and ACI and the main focus of the paper. I think this is not exactly what the authors want to show. In addition, the model version and modifications lacks some clear outlines. For example, WRF-Chem is first designed to simulate aerosol cycle, such as by MOSAIC; ASI is further included by coupling SNICAR (in CLM4) with aerosol cycles. Therefore, it would be better if this section can be re-organized as follows: brief description of model framework (WRF-Chem and WRF), representation of aerosol cycles, and aerosol effects (in the order of ARI, ACI, and ASI as in Introduction). Following the model description, some configurations for the specified simulation (such as domain, resolution, initial and boundary conditions, emission files, etc) in this study can be presented. Lines 195-223 can be kept as it is.

(4) Table 2, Lines 199-215: I am wondering what kinds of chemical species are transported into the domain. Do these species include dust or anthropogenic aerosols? Please explicitly mention this. If they include dust, NoDust should be NoLocDust. If they include anthropogenic aerosols, NoAnth should be NoLocAnth. Since their domain only covers a small region of Southwest United States, is it possible that dust and anthropogenic aerosols are also transported from adjacent regions (California-Arizona borders, Arizona, New Mexico, and the country of Mexico)? The authors only mention the long-range transportation from Asia and Africa. Please also clarify this.

(5) Lines 261-273: The evaluation of model simulations are only on the atmospheric aerosol. This study lacks the evaluation of aerosol-in-snow concentrations. Reasonable simulations of airborne aerosols don't necessarily imply reasonable simulation of aerosol-in-snow distribution, as there are lots of processes going after aerosol deposition on snow. Although the observations may be limited, some basic examination of aerosol-in-snow concentrations and their evaluation (if possible) is desirable to increase the reliability of ASI in this study. The results can be put in the supplement.

Specific comments:

Title: There is a word "convection-permitting" in title, but it is not mentioned in the main text. To increase the significance, I would suggest adding some brief discussions on the benefit of convection-permitting WRF-Chem simulations in Introduction.

Lines 35-36: Please make the order of ARI, ASI, and ACI consistently throughout the paper.

Lines 46-47: Transported anthropogenic aerosols or transported aerosols?

Line 50: Please mention the year for the period (since there is only a year for comparison).

Lines 70-71: The most adiabatic structure of the atmosphere is not clear.

Lines 87-88: The short atmospheric residence time can't cause geographical distributions. Compared to natural aerosols (dust), anthropogenic aerosols with smaller particles can be transported for a longer distance and a longer residence time. Please clarify.

Lines 191-192: Is the impact of aerosol on ice cloud formation included in the model?

Line 194: How long is the timestep?

Lines 197-199: If the results are similar, why are they still provided? Please clarify.

Line 222: Is NARS similar to the CTRL, except that ARI and ASI are not included?

Lines 237-238: Is it possible to find a reference for CPC? In addition, I cannot open the link for CPC data (Line 496).

Line 244: I am wondering how to get DWR data? What is the resolution? Is it gridded dataset or station measurement? It is not found in Data availability.

Line 245: Is it possible to find a reference for CIMIS? If so, please delete "http://www.cimis.water.ca.gov/", since Data availability is the place to mention it.

Lines 249-251: Does this affect both CPC and DRW datasets? Please clarify it.

Lines 291-293: what period is used for the calculation of difference and for daily data?

Lines 317-319: Probably mention that increase in temperature by reduced snow amount also overwhelms the decrease of temperature which may be caused by more clouds.

Line 322: I cannot find the runoff results..

Line 327: what's the aerosol-snow albedo feedback? Are you meaning snow-albedo feedback?

Lines 328-329: Please mention that reduced SWE can also initialize the snow albedo feedback.

Lines 347-348: The increased SWE can be canceled out to some extent by reduced snowfall (Lines 344-345). Please don't just mention the increased SWE and reduced snowfall separately, but consider them together (northern part of Sierra: ARI>ACI; southern part of Sierra: ACI>ARI).

Lines 358-359: Please be aware that this only applies to the total runoff change, but not to the monthly change which the snowmelt change also contributes to.

Lines 372-374: The authors are talking about the relative change here. Why is the relative change of runoff smaller when the relative change of SWE is larger? This can be partly explained by the slightly smaller change of precipitation (both liquid and solid form of precipitation are converted to runoff, soil water, and evapotranspiration eventually). Is it possible that the change of evapotranspiration also contributes?

Line 397: what's the orographic forcing?

Lines 423-424: The definition of surface runoff can be put earlier in Line 352 (when it appears at the first time).

Lines 425-426: If the authors are talking about total runoff (in an annual scale), surface runoff is mainly associated with precipitation. But in a monthly scale, surface runoff

is mainly associated with rainfall and snowmelt, and a portion of snowfall will become surface snow accumulation (epically for the winter season). In the melting season, precipitation is mainly in the terms of rainfall, which will mostly become runoff. Please clarify this.

Lines 428-430: Please indicate this is consistent with change of SWE.

Line 431: Please add "less snowpack available for melting caused by" before "earlier snowmelt".

Lines 462-463: Again, this is for longer time scale (e.g., annual). In a shorter time scale, runoff can be generated from snowmelt. This is actually one point in this study: seasonal cycle of runoff is modified by aerosols through the impacts of aerosol on snowpack.

Line 467: Probably add "less snowpack available for melting caused by" before "earlier snowmelt". In the earlier period of snowmelt, the author can say there is more runoff due to earlier snowmelt. But in the late period of snowmelt, it is more correct to say that less runoff is due to less snowpack available for melting to generate runoff.

Lines 481-486: Does underestimation of AOD imply that the aerosol effects are also biased low here? If so, please explicitly mention it.

Lines 489-492: The authors have mentioned that aerosol effect on ice cloud formation is not explicitly treated in the model (Line 314). They also mentioned the potential significance of aerosol effect on snow formation (Lines 122-124). May the limitation of the model (i.e., inexplicit treatment of aerosol effect on ice cloud formation) affect the results presented here? It will be helpful to add a brief discussion.

Figures: Surface runoff is one of key variables the authors focus on. However, the authors don't present any spatial distribution and temporal evolution as other variables (precipitation, SWE, T2). I would suggest adding the spatial distribution and temporal evolution of runoff as well as spatial distribution of runoff change by aerosols. They

can be put in supplement.

Figure 1: If possible, please provide some indicators for the main mountains (including Sierra Nevada and Klamath Mountains) and valleys, which can be easily referred to in the main text. This will help the general readers of the journal.

Figure 3 captions, Lines 791-794: I would say "from CTRL simulations and xxx observations" instead of "simulated from CTRL and the observations from xxx". In addition, do (a) and (c) refer to a regional mean? Please clarify.

Figure 3: X-axis in (c) is overlaid by white shaded box.

Figure 5: I am wondering how the authors do the significant test, as there is only one year simulation for each experiment.

Figures 6-12: Can the result of significant test be shown as in Figure 5? This is normally required as the authors mention multiple times of "significant" in the text (Lines 304, 313, 317, 326, 339, 369, 479).

Figures 14-17: Please add the "zero" line in the figures for easy viewing.
* * *

---

## Referee Comment (RC2) · Anonymous Referee #2 · 21 Nov 2017

**Review of Wu et al., ACP, 2017 - Impacts of Aerosols on Seasonal Precipitation and Snowpack in California Based on Convection-Permitting WRF-Chem Simulations**

This paper uses the WRF-Chem regional model at 4km resolution to attempt to diagnose the effects of aerosols from different sources upon temperature, precipitation, snowfall and cloud properties over the California region. Simulations are run for 10 months for two different years.

There are some interesting results, but there are also some issues that need addressing before publication. My main concern is whether the "CLEAN" low aerosol case has too few aerosols (see below), which would lead to overestimates of the aerosol effect. But there are numerous others listed below. There are also a number of grammatical mistakes – I picked out a few, but there are more. Hopefully these will be picked up by the proof reader.

**Overall comments**

Model setup – I'm a bit confused by the CLEAN case. Do you set all the lateral boundaries to zero for all aerosols? Or just anthropogenic ones? If it is all aerosols and there are no local sources then I would imagine this would soon lead to there being very little or no aerosol at all in the domain (local non-anthropogenic aerosol only?)? If so, then what does the model do in zero aerosol situations in terms of droplet activation (since this may be the case for regions near the inflow boundary)? It would make more sense to allow non-anthropogenic aerosols into the lateral boundaries, so that what comes in is more like a clean background case. Or is this what has been done? It should be made clear in the manuscript.

There is a comparison of the model to observations in terms of the meteorology, but not for the aerosl properties. Since this is key to the results, it would be good to give some details of the comparison of the aerosol properties to observations rather than referring to the previous paper.

It would be good to mark/list the observational sites that are used.

It mentions that there is no effect of aerosol upon ice in the model - can you discuss the potential impact of this? E.g., more aerosol might lead to more ice nucleating particles, which could affect snowfall/ice production, etc. Perhaps a sensitivity test could be done whereby the number of ice nucleating particles (INP) are enhanced. Is an INP scheme used, and if so which one?

Do the precipitation rates that are quoted include ice phase precipitation or just liquid? It would be helpful to try to separate the liquid and ice phase precipitation.

Is it really the case that the transported aerosol comes from East Asia rather than more local sources? E.g. there seems to be a region of high AOD in Fig. 4d close to where Los Angeles is. Since the

transported aerosol seems to be one of the biggest contributors the source regions for this should be examined more carefully. Wind arrows showing the mean flow are also needed for Fig. 4 (or Fig. 1).

What causes the fairly large increases in SWE NW of the mountains?

It would be good to comment on the fact that the anth+dust+tran effects do not seem to add up to total effects – i.e., the overall combined effect seems to be greater than the sum of the parts.

**Line-by-line comments**

Abstract – you should mention the study period before you start to talk about the results.

L37 – "snow water equivalent (SWE)," – it is never explained what is meant by this. It sounds like it is the accumulated amount of snow that has fallen to the surface expressed as mm of water equivalent. But over the time period is never given. Presumably it is over the whole study period? This should be explained more thoroughly in the text before it is used.

L238 – Does the CPC rain rate product include only rain (and not snow)? This should be mentioned for clarity.

L245 – "For SWE, daily 245 mean SWE simulations are compared with measurements collected at Snow Telemetry" – should this be daily accumulated measurements rather than a mean?

L251 – "Model data are sampled onto observational sites before the comparison is conducted" – This information needs to come before the results are discussed (and put in the caption too). Does it apply to all of the observational data? Where are the observational sites? They should be listed or marked on the map, or at least some information on how many there are and their distribution, etc.

L258 – "Therefore, the WRF-Chem model that we employ in this study is a reliable tool for examining the impact of aerosols on the seasonal variations of 259 precipitation and snowpack in California, especially over the Sierra Nevada"

   The results show a good representation of the meteorology and precipitation/snow, but it is a bit of an extrapolation to say that this means that it can reliably be used for aerosol-cloud interactions. E.g. we don't know how well it captures the aerosol and how its interaction with clouds. Better to say that the model represents the meteorology in a realistic manner. Or move the sentence to after you have explained how WRF compares for aerosol in the next paragraph.

L283 – "Transported aerosols, including dust and biological aerosols from East Asia 283 (Creamean et al., 2013), are carried into the domain by atmospheric circulation and widely 284 distributed, with more over the central valley due to the trapping of aerosols by the surrounding 285 mountains (Fig. 4d)."

   Is it really the case that the transported aerosol comes from East Asia rather than more local sources? E.g. there seems to be a region of high AOD in Fig. 4d close to where Los Angeles is. Since the

transported aerosol seems to be one of the biggest contributors the source regions for this should be examined more carefully.

Also, can you explain how you made these plots? E.g. are they from runs with just the particular emissions included (anth, dust, trans), or did you have to do some differencing between the CTRL case and the e.g. no transport simulation?

L305 – you don't talk about the effect on SWE here even though it appears stronger than for the ARI where you did discuss it.

L318 - can you elaborate on why there is less SWE due to ACIs? What is the proposed mechanism and do you have evidence for it? Is it related to their being less liquid precipitation (e.g. less raindrop freezing, smaller droplets and so less droplet freezing)? Or does precipitation here include that from snow/ice? It might be argued that the higher LWPs might allow more liquid water to become frozen giving more SWE. Later on (L408) you say that the extra clouds from the ACI effect lead to less surface melt and more SWE for the lower elevation regions – can you explain/show whether the precipitation (or other) effect dominate over the temperature effect for the mountain tops, but not the lower elevations?

Likewise, can you please elaborate on why the albedo decreases and why the surface temperature increases. Is it due to the lack of fresh snow so that there is more exposed aged snow (although , or perhaps there are regions with no snow at all (at the start of the season perhaps)?

L343 – "It is shown that 343 transported aerosols also reduce the precipitation through ACI (Fig. 12a),"

L432 – "the impact of 432 aerosols is to speed up snowmelt at mountain tops." – This sentence should be removed since it suggests that aerosol enhance overall snowmelt when actually they reduce the runoff overall. There is a small effect of speeding up the onset, but this has already been mentioned and does not need to be said again since it ignores the snowmelt reduction effect (through the precipitation decrease).

Conclusions/L441 – "Temperature: Dust aerosols warm the mountain top surfaces through ASI (0.12 K)," – would be good to say that the numbers in brackets are domain mean changes. Also, you should re-iterated the abbreviations ASI, etc. in the text at the start of the conclusions and refer to Table 4.

L468 – "Therefore, one of the important impacts of aerosols is to speed up the snowmelt at 468 mountain tops." Is this really one of the most important aspects? Since the effect on runoff then goes on to be dominated by the reduction in the precipitation. And you can't be sure how much effect the earlier snow melt is having on that – most of the effect could be coming from the precip reduction?

**Tables/Figures**

Table 3 – perhaps it is worth mentioning that these experiments use the CTRL aerosol emissions.

Fig. 1 – It would be useful to label the valley, big cities and other regions of interest in Fig. 1. Also, the colorbar is a bit strange since the colors around 150m and 600m seem to repeat.

Fig. 2 – it is confusing to say that the SWE is averaged over the time period since presumably it is the accumulated snow amount?

Fig.3 – should state the region being considered here and in the text – is it the whole model domain? It would be good to also use a dashed line for the model to help distinguish it for colorblind readers.

**Typos**

L230 – "in CTRL experiment" -> "in the CTRL experiment"

L233 - "in the northern California" -> "in northern California"

L235 – "while colder temperature is found" -> "while colder temperatures are is found"

L314 - "because aerosol effect" -> "because the aerosol effect"

L316 - "associated with ACI effect" -> "associated with the ACI effect"

L358 – "contributes to the increase (1.88%)." – "contributes to an increase (1.88%)." (since overall there is a decrease).

L484 – "importance" -> "important"

---

## Author Comment (AC2) · 16 Feb 2018

We would like to thank the Reviewers for their constructive comments and suggestions for the improvement of our manuscript. We have carefully revised the manuscript following these comments and suggestions. Below we have listed the referees' comments in black and our response in blue.

**Reviewer 1**

Aerosols can induce large impacts on the regional climate and hydrologic cycles. Currently the aerosol effects are still not well understood, especially for the individual and combined effects of different underlying mechanisms (direct, indirect, and feedback).

This study presents a comparison of different aerosol effects including aerosol radiation interaction (ARI), aerosol-cloud interaction (ACI), and aerosol-snow interaction (ASI) on the regional climate in California based on WRF-Chem simulations. The study also shows the different effects induced by local dust emissions, local anthropogenic emissions, and transportation. Overall, the manuscript is well written, and most of the content is well organized. The scientific findings are significant to our understanding of climatic effects of different aerosols. This study is useful for the relevant research community on unraveling the aerosol affects in climate and hydrologic cycles.

However, some statements are not clear and some of them may need further evidence. Part of the manuscript can be better organized for easy following. I have some suggestions and comments that I would like the authors to consider before the manuscript can be accepted for publication in ACP.

Response: We appreciate the reviewer's valuable comments. We have addressed these comments in the revised manuscript. Point-to-point responses are given below.

**Major comments:**

(1)     Lines 254-256, Figure 3, Lines 36-40 (Abstract): The authors states that the model simulations represent reasonable magnitude of SWE, because SNOTEL data underestimates real SWE. They deduce the underestimate of SNOTEL SWE from "The main issue with weighing-type gauges for snowfall estimation is the undercatch of approximately 10%–15% due to wind (Serreze et al., 2001; Yang et al., 1998; Rasmussen et al., 2001). " (Lines 249-251). I should mention that snowfall is not SWE. They are measured differently: snowfall referring to a solid form of precipitation is measured by gauges, while SWE is measured using a snow pillow (https://www.wcc.nrcs.usda.gov/about/mon_automate.html). Therefore, underestimation of snowfall doesn't mean underestimation of SWE. If SNOTEL SWE is not underestimated compared to the reality, the model (with aerosol effects) may have large biases in SWE (up to~100 mm) (Figure 3b).

Response: We agree with the reviewer that the underestimation of snowfall does not mean an underestimation of SWE. We revised the text as following (lines 294-300):

For SWE, daily mean SWE simulations are compared with measurements collected at Snow Telemetry (SNOTEL) stations. SWE is measured using a snow pillow sensor and biases in SWE measurement could occur when temperature differences between surrounding ground cover and the pillow sensor create uneven distribution of snow (Meyer et al., 2012). Both under- and over-estimation could happen depending on the snowmelt conditions and the snow density rate of change (Serreze et al., 1999; Serreze et al., 2001; Johnson and Marks, 2004).

The authors state that inclusion of aerosol effects reduce the model biases (Abstract). Although it is generally true, it is not simply the case for a model simulation regarding the large uncertainties in current models. With aerosol effects, WRF-Chem reduces SWE biases by 0-60 mm (Figure 15), but still has the bias of ~100 mm (mentioned above, if SNOTEL SWE is not biased low). Although the authors can still get the conclusion of reduction of SWE biases with aerosol effects, discussion on other reasons for the model biases (potentially larger than the biases that can be reduced by including aerosol effects) is desirable and helpful.

Response: Discussion on other reasons for the model biases are included in the text (lines 584-586):

Our model simulation produces relative larger SWE than the SNOTEL observations. Improvement of snowpack simulation in the land surface model is needed for accurate quantification of aerosol impacts on snowpack.

In addition, model simulations are not always improved with the inclusion of aerosols effects. For example, CTRL simulation underestimates precipitation in April (Figure 3a). If the aerosol effects are removed, simulated precipitation is larger (Figure 14), which is more consistent with the observation.

Response: In the abstract, we talk about general performance of the model simulation with aerosol effects included. We agree with the reviewer that the model simulations are not always improved with the inclusion of aerosol effects in all months. The different performance of the model simulation in different months is clarified in the main text as following (lines 303-305).

In the relative dry months from February to June, the simulated precipitation has similar magnitude to the observations, with slightly overestimation or underestimation in different months.

For precipitation and temperature, there are multiple observations available for comparison with the model simulations. Without the investigations of the reliability of each observation, the selected observations may be arbitrary. Besides the CPC, DWR, and CIMIS observations used in this study, there are also other datasets (including a widely-used dataset, PRISM-Parameter-elevation Regression Independent Slopes Model) available but not included. The resolution of PRISM (4 km), much higher than CPC (0.25 degree) used, is also similar to the model resolution (4km). I am wondering how the simulation results are compared to the PRISM observation at similar resolution.

Response: Thanks for the suggestion. We have added comparison with PRISM in the revised manuscript. As shown in the following Figure 1, the CTRL simulation has better agreement with PRISM than CPC. As pointed out by the reviewer, the PRISM data is widely used and has the same resolution as the CTRL run. Thus, we replace CPC by PRISM in Fig. 2f, while including both CPC and PRISM in Fig. 3a in the revised manuscript. The text is revised accordingly.

[Figure]

Figure 1. Mean precipitation (mm day$^{-1}$) from (left panel) CTRL, (middle panel) PRISM, and (right panel) CPC.

Overall, more investigation is needed to support the improvement of model performance when aerosol effects are included, by comparison of model results with more observation datasets and consideration of the reliability of these observations.

Response: Following the reviewers' comments, we have added more comparisons to support the improvement of model performance when aerosol effects are included. We have added comparisons with PRISM in Fig. 2 and Fig. 3a. The comparison of AOD between the MISR observation and the CTRL simulation is included in Fig. 4a and 4b. We have also added a figure in the supplementary information (Fig. S2) to evaluate the model simulations of snow albedo which is related to the direct effect of ASI.

(2) Table 3, Lines 216-223: The authors decompose the effects of ARI, ACI, and ASI from these multiple experiment. Do they assume the linear combination of ARI, ACI, and ASI? It is possible that ARI, ACI, and ASI can be interacted to generate overall effects. CTRL-NARI (CTRL-NASI) may include the interaction of ARI/ACI and ARI/ASI (ARI/ASI and ACI/ASI), which may be different from NASI-NARS (NARI-NARS). If any difference between CTRL-NARI and NASI-NARS (CTRL-NASI and NARI-NARS) is found, it is also helpful if the authors can explicitly

mention this nonlinear combination of ARI/ACI/ASI. Although it is difficult to identify the interaction of ARI, ACI, and ASI, at least some discussions are needed.

Response: We agree with the reviewer that the overall aerosols effects are not a linear combination of the ARI, ASI, and ACI effects. Following the reviewer's suggestion, we have added the following discussion in the revised paper (lines 224-230):

Since the model explicitly considers different sources and types of aerosols and contains the physical processes to represent various aerosol effects (ARI, ASI, and ACI), it is useful to decompose the aerosol effects based on aerosol sources/types and pathways. Note that the overall aerosols effects are not a simple sum of different aerosol sources/types, nor a linear combination of the ARI, ASI, and ACI effects. Differences between various simulations, however, help to identify the effect of a single source or pathway and the decomposition approach is a common practice in the experiment design of modeling studies.

(3) Section2: The authors describe the three pathways of aerosol effects in the order of ARI, ACI, and ASI in Introduction, but describe their representation in WRF-Chem in the order of ASI, ARI, and ACI in Section 2. This tends to give the readers an impression that ASI is more important than ARI and ACI and the main focus of the paper. I think this is not exactly what the authors want to show. In addition, the model version and modifications lacks some clear outlines. For example, WRF-Chem is first designed to simulate aerosol cycle, such as by MOSAIC; ASI is further included by coupling SNICAR (in CLM4) with aerosol cycles. Therefore, it would be better if this section can be re-organized as follows: brief description of model framework (WRF-Chem and WRF), representation of aerosol cycles, and aerosol effects (in the order of ARI, ACI, and ASI as in Introduction). Following the model description, some configurations for the specified simulation (such as domain, resolution, initial and boundary conditions, emission files, etc) in this study can be presented. Lines 195-223 can be kept as it is.

Response: We appreciate the reviewer's suggestion on the structure of the manuscript. The ASI pathway included in our WRF-Chem version is the major difference from the public released version. Thus we put ASI in the first order with more detailed description. ARI and ACI have been documented by many papers, such as Fast et al. (2012, 2014) and Zhao et al. (2010, 201, 2013a, 2013b). Therefore, we give a brief introduction of ARI and ACI following ASI.

(4) Table 2, Lines 199-215: I am wondering what kinds of chemical species are transported into the domain. Do these species include dust or anthropogenic aerosols? Please explicitly mention this. If they include dust, NoDust should be NoLocDust. If they include anthropogenic aerosols, NoAnth should be NoLocAnth. Since their domain only covers a small region of Southwest United States, is it possible that dust and anthropogenic aerosols are also transported from adjacent regions (California-Arizona borders, Arizona, New Mexico, and the country of Mexico)? The authors only mention the long-range transportation from Asia and Africa. Please also clarify this.

Response: The initial and boundary chemical conditions are taken from the MOZART-4 global chemical transport model. The chemical species transported into the model domain include organic carbon, black carbon, sulfate, nitrate, ammonium, sea salt, dust, etc.. Following the reviewer's suggestion, we give a brief description of the chemical species which are transported into the domain, including dust and anthropogenic aerosols. The transported aerosols investigated in this study refer to the aerosols transported from outside the model domain, including those from East Asia and other regions. It is clarified in the revised manuscript (lines 241-247).

In the NoDust and NoLocAnth experiments, only local dust or local anthropogenic aerosols are excluded. We have followed the reviewer's suggestion and change the experiment name to NoLocDust and NoLocAnth in the revised manuscript.

(5) Lines 261-273: The evaluation of model simulations are only on the atmospheric aerosol. This study lacks the evaluation of aerosol-in-snow concentrations. Reasonable simulations of airborne aerosols don't necessarily imply reasonable simulation of aerosol-in-snow distribution, as there are lots of processes going after aerosol deposition on snow. Although the observations may be limited, some basic examination of aerosol-in-snow concentrations and their evaluation (if possible) is desirable to increase the reliability of ASI in this study. The results can be put in the supplement.

Response: Since the observations on aerosol-in-snow concentrations are rather limited both spatially and temporally as the reviewer pointed out, it's very difficult to conduct direct comparisons with model simulations. Following the reviewer's suggestion, we instead added a figure in the supplementary information to evaluate the model simulations of snow albedo which is directly affected by the ASI (Fig. S2). The model simulated snow albedo is compared with the product from NASA Land Data Assimilation Systems (NLDAS) Mosaic (MOS). It is shown that model simulation provides rather reasonable estimate of the snow albedo with ASI included (Fig. S2 and the following Figure 2, lines 339-344).

[Figure]

Figure 2. Spatial distribution of surface albedo averaged over October 2012 to June 2013 from (a) NLDAS data assimilation and (b) CTRL simulation.

**Specific comments:**

Title: There is a word "convection-permitting" in title, but it is not mentioned in the main text. To increase the significance, I would suggest adding some brief discussions on the benefit of convection-permitting WRF-Chem simulations in Introduction.

Response: The reviewer's comment is well taken. A brief discussion on the benefit of convective-permitting WRF-Chem simulations has been added in the revision (lines 205-223):

One important subgrid process in climate models is the representation of deep convection. Parameterizing deep convection is challenging and the use of convection parameterization schemes leads to common errors such as misrepresentation of the diurnal cycle of convective precipitation (e.g., Dai et al., 1999; Brockhaus et al., 2008), underestimation of dry days (e.g., Bergetal., 2013) and precipitation intensity (e.g., Prein et al., 2013; Fosser et al., 2014; Ban et al., 2014), and overestimation of low-precipitation frequency (e.g., Bergetal., 2013). Although recently developed parameterization schemes lead to improvements in the simulation of precipitation intensity (Donner et al., 2011), intraseasonal variability (Benedict et al., 2013), and diurnal cycles (Bechtold et al., 2014), a promising remedy to the error-prone model simulations using convective parameterizations is the use of convection-permitting model with horizontal grid spacing of about 4 km or less (e.g., Satoh et al., 2008; Prein et al., 2013; Ban et al., 2014). Advances in high-performance computing allowed refinement of the model grids well below 10 km. At these

scales, convection parameterization schemes may be switched off as deep convection starts to be resolved explicitly (e.g., Weisman et al., 1997). According to Prein et al. (2014), it seems prudent to use horizontal grid spacing of 4 km or less for convection-permitting model simulations. The 4 km simulation can also represent topography and inhomogeneous distribution of anthropogenic emission and precipitation better, leading to a better representation of aerosol distribution comparing to the 20 km simulation (Wu et al., 2017).

Lines 35-36: Please make the order of ARI, ASI, and ACI consistently throughout the paper.

Response: The order is kept in the revision.

Lines 46-47: Transported anthropogenic aerosols or transported aerosols?

Response: Here it means transported aerosols. We have change it to "Transported aerosols and local anthropogenic aerosols" (line 46).

Line 50: Please mention the year for the period (since there is only a year for comparison).

Response: Years have been added as "from October 2012 to June 2013" (line 50).

Lines 70-71: The most (moist) adiabatic structure of the atmosphere is not clear.

Response: Following the reviewer's comment, we have revised the following sentence (lines 69-76):

Previous studies suggested that warming trends are amplified in mountains compared to lowlands (Pepin et al., 2015). The amplified warming in mountain areas, also referred to as elevation-dependent warming, is generally attributed to a few important processes (Pepin et al., 2015), such as water vapor changes and latent heat release, surface water vapor changes, radiative flux changes associated with three-dimensional rugged topography (Gu et al., 2012a; Liou et al., 2013; Lee et al., 2015; Zhao et al., 2016), and snow-albedo feedback (Leung et al., 2004). A review and assessment of the mechanisms contributing to an enhanced warming over mountain areas is given in Pepin et al. (2015).

Lines 87-88: The short atmospheric residence time can't cause geographical distributions. Compared to natural aerosols (dust), anthropogenic aerosols with smaller particles can be transported for a longer distance and a longer residence time. Please clarify.

Response: The local geographical distributions of anthropogenic aerosols over California is also related to another reason: the regional topography. Following the reviewer's comment, we have revised this sentence (lines 92-95).

"Anthropogenic aerosols are geographically distributed because of localized emission sources, the short atmospheric residence time, and regional topography. With valleys and surround mountain

barriers, dispersion of air pollutants is more difficult for locally emitted anthropogenic air pollution."

Lines 191-192: Is the impact of aerosol on ice cloud formation included in the model?

Response: The impact of aerosols on ice cloud is not included in the model, and therefore there no significant changes in ice water path (IWP, Figs. 8b & 8d). This has been clarified in the model description part (lines 178-179) and results part (lines 378-379) of the original manuscript.

Line 194: How long is the timestep?

Response: The time step is 20 seconds and has been added in the revision (line 199).

Lines 197-199: If the results are similar, why are they still provided? Please clarify.

Response: It is clarified as the following (lines 202-204).

To test the robustness of the results, simulations are also conducted for year 2013-2014, and similar results are found. In the following section, our analysis focuses on year 2012-2013, while quantitative information of the aerosol impacts for year 2013-2014 is provided for comparison.

Line 222: Is NARS similar to the CTRL, except that ARI and ASI are not included?

Response: Yes, NARS is similar to the CTRL, except that both ARI and ASI are not included. We have rephrase the sentence (line 263).

Lines 237-238: Is it possible to find a reference for CPC? In addition, I cannot open the link for CPC data (Line 496).

Response: The link for the CPC data is updated:
https://www.esrl.noaa.gov/psd/data/gridded/data.unified.daily.conus.html

The following reference for CPC has been added in the revision (line 292; 647-649):
Chen, M., Xie, P., and Co-authors: CPC Unified Gauge-based Analysis of Global Daily Precipitation, Western Pacific Geophysics Meeting, Cairns, Australia, 29 July - 1 August, 2008.

Line 244: I am wondering how to get DWR data? What is the resolution? Is it gridded dataset or station measurement? It is not found in Data availability.

Response: DWR data can be downloaded at http://cdec.water.ca.gov/snow_rain.html. It is station measurement. We added the link in the Data Availability part (line 594).

Line 245: Is it possible to find a reference for CIMIS? If so, please delete "http://www.cimis.water.ca.gov/", since Data availability is the place to mention it.

Response: The following reference for CIMIS has been added in the revision (line 293; lines 833-834):
Snyder, R. L.: California irrigation management information system. Am. J. Potato Res. 61(4): 229–234, 1984.

Lines 249-251: Does this affect both CPC and DRW datasets? Please clarify it.

Response: This statement is removed in the revision. We are not aware of any study that investigated wind effects on CPC or DWR datasets.

Lines 291-293: what period is used for the calculation of difference and for daily data?

Response: The differences are averaged over October 2012 to June 2013 for the contour maps. This information has been added in the revision (line 355).

Lines 317-319: Probably mention that increase in temperature by reduced snow amount also overwhelms the decrease of temperature which may be caused by more clouds.

Response: The reviewer's suggestion has been well taken. The text has been revised accordingly (lines 381-384).

Line 322: I cannot find the runoff results.

Response: The runoff results are added in Supplementary Information, Fig. S3. (line 387).

Line 327: what's the aerosol-snow albedo feedback? Are you meaning snow-albedo feedback?

Response: Here it means the aerosol induced snow-albedo feedback. The text has been revised (lines 392).

Lines 328-329: Please mention that reduced SWE can also initialize the snow albedo feedback.

Response: We appreciate the reviewer's suggestion. We have revised the sentence as: "For the ACI effect, however, warming over the mountain region is a result from the reduced SWE which can also induce snow-albedo feedback and result in smaller surface albedo and more surface absorption of solar radiation." (lines 393-395).

Lines 347-348: The increased SWE can be canceled out to some extent by reduced snowfall (Lines 344-345). Please don't just mention the increased SWE and reduced snowfall separately, but consider them together (northern part of Sierra: ARI>ACI; southern part of Sierra: ACI>ARI).

Response: Following the reviewer's comment, the text has been revised as (lines 409-415):

It is shown that transported aerosols also reduce the precipitation through ACI (Fig. 12a), which exceeds the ARI effect and leads to decreased SWE and increased temperature over the southern part of Sierra Nevada (Figs. 12b and 12c). Over the central valley, as well as over the northern part

of the Sierra, temperature decreases (Fig. 12c) due to the relatively larger ARI effect of the transported aerosols compared to ACI, resulting in less snowmelt and increased SWE over that region (Fig. 12b).

Lines 358-359: Please be aware that this only applies to the total runoff change, but not to the monthly change which the snowmelt change also contributes to.

Response: We agree with the reviewer that snowmelt change also contributes to the change in runoff. We revised the sentence as (lines 426-428):

Overall changes in surface runoff are similar to those in precipitation, accompanied by contributions from changes in snowmelt.

Lines 372-374: The authors are talking about the relative change here. Why is the relative change of runoff smaller when the relative change of SWE is larger? This can be partly explained by the slightly smaller change of precipitation (both liquid and solid form of precipitation are converted to runoff, soil water, and evapotranspiration eventually). Is it possible that the change of evapotranspiration also contributes?

Response: The relative change of surface runoff at the mountain tops in year 2013-2014 is smaller than year 2012-2013 because the mean surface runoff in year 2013-2014 (0.33 mm day$^{-1}$) is larger than that in year 2012-2013 (0.27 mm day$^{-1}$), possibly contributed by less SWE and faster snowmelt at the mountain tops in year 2013-2014. The corresponding changes in evapotranspiration are −0.12% in year 2012-2013 and −1.20% in year 2013-2014, respectively, which also contributes to the relatively smaller change of surface runoff in year 2013-2014 at the mountain tops.

We have added this in the revision (lines 441-447).

Line 397: what's the orographic forcing?

Response: Here we mean "precipitation due to orographic forcing". We have reworded it as "the orographic precipitation over the mountain region". Orographic lift occurs when an air mass is forced from a low elevation to a higher elevation as it moves over rising terrain. As the air mass gains altitude it quickly cools down adiabatically, which can raise the relative humidity to 100% and create clouds and, under the right conditions, precipitation. Orographic forcing is an efficient and dominant mechanism for harnessing water vapor into consumable freshwater in the form of precipitation, snowpack, and runoff. It has been estimated that about 60–90% of water resources originate from mountains worldwide, including the western slope of the Sierra Nevada range in California.

Lines 423-424: The definition of surface runoff can be put earlier in Line 352 (when it appears at the first time).

Response: We appreciate the reviewer's suggestion. The definition of surface runoff has been moved earlier when the overall changes in surface runoff are discussed (lines 425-426).

Lines 425-426: If the authors are talking about total runoff (in an annual scale), surface runoff is mainly associated with precipitation. But in a monthly scale, surface runoff is mainly associated with rainfall and snowmelt, and a portion of snowfall will become surface snow accumulation (epically for the winter season). In the melting season, precipitation is mainly in the terms of rainfall, which will mostly become runoff. Please clarify this.

Response: We agree with the reviewer that snowmelt plays an important role in surface runoff. We have revised the text following the reviewer's comment (lines 496-508):

For lower elevations where there is not much snow, surface runoff is mainly associated with precipitation and the changes present a similar pattern to those in precipitation (Fig. 17c). Changes in surface runoff for the whole area present similar patterns to those of the lower elevations because of the larger area of lower elevations (Fig. 17a). However, for mountain tops, changes in surface runoff are also associated with changes in snowmelt. Surface runoff over mountain tops shows a slight increase in spring, and then a decrease after April (Fig. 17b). The increase can be explained by the effect of dust aerosols deposited on the snow, which reduces the snow albedo through ASI and warms the surface, leading to more and earlier snowmelt than normal, consistent with negative changes in SWE. The decrease after April is a combined effect of less snowpack available for melting caused by earlier snowmelt due to dust aerosols and reduced precipitation caused by transported and anthropogenic aerosols through ACI. Thus, the impact of aerosols is to speed up snowmelt at mountain tops in spring and modify the seasonal cycle of surface runoff.

Lines 428-430: Please indicate this is consistent with change of SWE.

Response: Done (line 504).

Line 431: Please add "less snowpack available for melting caused by" before "earlier snowmelt".

Response: Done (line 505).

Lines 462-463: Again, this is for longer time scale (e.g., annual). In a shorter time scale, runoff can be generated from snowmelt. This is actually one point in this study: seasonal cycle of runoff is modified by aerosols through the impacts of aerosol on snowpack.

Response: We really appreciate the reviewer's comment. We have added the effect of snowmelt in monthly variations (lines 539-540; 546).

Line 467: Probably add "less snowpack available for melting caused by" before "earlier snowmelt". In the earlier period of snowmelt, the author can say there is more runoff due to earlier snowmelt. But in the late period of snowmelt, it is more correct to say that less runoff is due to less snowpack available for melting to generate runoff.

Response: Done (lines 543-544).

Lines 481-486: Does underestimation of AOD imply that the aerosol effects are also biased low here? If so, please explicitly mention it.

Response: The reviewer has a very good point. We have added in the revision: "The underestimate of AOD in the model implies that the simulated aerosol effects could also be biased low." (lines 562-563).

Lines 489-492: The authors have mentioned that aerosol effect on ice cloud formation is not explicitly treated in the model (Line 314). They also mentioned the potential significance of aerosol effect on snow formation (Lines 122-124). May the limitation of the model (i.e., inexplicit treatment of aerosol effect on ice cloud formation) affect the results presented here? It will be helpful to add a brief discussion.

Response: In the current WRF-Chem model, the aerosol effect on ice clouds is not included. ACI associated with ice clouds are more complex than that with liquid clouds. For example, a few studies have shown that negative Twomey effects may occur with aerosols and ice clouds, in which increased aerosols (and thus ice nuclei) lead to enhanced heterogeneous nucleation that is associated with larger and fewer ice crystals as compared to the homogeneous nucleation counterpart (DeMott et al., 2010; Chylek et al., 2006, Zhao et al. 2018). A recent study shows that the responses of ice crystal effective radius to aerosol loadings are modulated by water vapor amount in conjunction with several other meteorological parameters. While there is a significant negative correlation between ice effective radius and aerosol loading in moist conditions, consistent with the "Twomey effect" for liquid clouds, a strong positive correlation between the two occurs in dry conditions (Zhao et al. 2018). Despite numerous studies about the impact of aerosols on ice clouds, the role of anthropogenic aerosols in ice processes, especially over polluted regions, remains a challenging scientific issue. The effect of anthropogenic aerosols on ice formation and cloud radiative properties may be a critical pathway through which anthropogenic activities affect regional climate and present the opportunities for further studies using observations and models.

Following the Reviewer's comment, we have added the above discussion about the possible influence of the INP effect in the revised manuscript (lines 568-583).

Figures: Surface runoff is one of key variables the authors focus on. However, the authors don't present any spatial distribution and temporal evolution as other variables (precipitation, SWE, T2). I would suggest adding the spatial distribution and temporal evolution of runoff as well as spatial distribution of runoff change by aerosols. They can be put in supplement.

Response: The spatial and temporal distribution of surface runoff is included in the Figures S1, S3 and S4 in the Supplementary Information of the revised manuscript.

Figure 1: If possible, please provide some indicators for the main mountains (including Sierra Nevada and Klamath Mountains) and valleys, which can be easily referred to in the main text. This will help the general readers of the journal.

Response: Following the reviewer's suggestion, the indicators for Sierra Nevada and Klamath Mountains have been provided in Fig. 1.

Figure 3 captions, Lines 791-794: I would say "from CTRL simulations and xxx observations" instead of "simulated from CTRL and the observations from xxx". In addition, do (a) and (c) refer to a regional mean? Please clarify.

Response: Captions have been modified following the reviewer's suggestion. All the data refer to an average for the stations used.

Figure 3: X-axis in (c) is overlaid by white shaded box.

Response: Changed.

Figure 5: I am wondering how the authors do the significant test, as there is only one year simulation for each experiment.

Response: The two-tailed Student's t test, in which deviations of the estimated parameter in either direction are considered theoretically possible, is applied to the 3-hourly data for each experiment in this study to measure the statistical significance of the sensitivity simulations (lines 352-355).

Figures 6-12: Can the result of significant test be shown as in Figure 5? This is normally required as the authors mention multiple times of "significant" in the text (Lines 304, 313, 317, 326, 339, 369, 479).

Response: The figures with the result of significant test look quite noisy. So we don't show the dots as in Fig. 5. For Figures 6-12, most of the data are statistically significant at a significance level of 70%. We added this explanation in the text (lines 362-363).

Figures 14-17: Please add the "zero" line in the figures for easy viewing.

Response: Done.

---

## Author Comment (AC3) · 16 Feb 2018

We would like to thank the Reviewers for their constructive comments and suggestions for the improvement of our manuscript. We have carefully revised the manuscript following these comments and suggestions. Below we have listed the referees' comments in black and our response in blue.

**Reviewer 2**

This paper uses the WRF-Chem regional model at 4km resolution to attempt to diagnose the effects of aerosols from different sources upon temperature, precipitation, snowfall and cloud properties over the California region. Simulations are run for 10 months for two different years.

There are some interesting results, but there are also some issues that need addressing before publication. My main concern is whether the "CLEAN" low aerosol case has too few aerosols (see below), which would lead to overestimates of the aerosol effect. But there are numerous others listed below. There are also a number of grammatical mistakes – I picked out a few, but there are more. Hopefully these will be picked up by the proof reader.

Response: We appreciate the reviewer's valuable comments. We have addressed these comments in the revised manuscript. Point-to-point responses are given below. We have done our best to correct grammatical mistakes.

**Overall comments**

Model setup – I'm a bit confused by the CLEAN case. Do you set all the lateral boundaries to zero for all aerosols? Or just anthropogenic ones? If it is all aerosols and there are no local sources then I would imagine this would soon lead to there being very little or no aerosol at all in the domain (local nonanthropogenic aerosol only?)? If so, then what does the model do in zero aerosol situations in terms of droplet activation (since this may be the case for regions near the inflow boundary)? It would make more sense to allow non-anthropogenic aerosols into the lateral boundaries, so that what comes in is more like a clean background case. Or is this what has been done? It should be made clear in the manuscript.

Response: In the CLEAN case, we set all the lateral boundaries to zero for all aerosols, while we keep all the transported chemical species. Aerosols are low in the simulation, but not zero, possibly due to aerosol chemistry. The CCN concentration at supersaturation of 0.1% is on the order of 10 $cm^{-3}$ at most time of the CLEAN simulation. The distribution of liquid water path and ice water path in the CLEAN simulation is also similar to that in the CTRL simulation, with differences in magnitude. So we think it is reasonable to use this setting to represent a clean background case. It is clarified in the manuscript (lines 248-254).

There is a comparison of the model to observations in terms of the meteorology, but not for the aerosol properties. Since this is key to the results, it would be good to give some details of the comparison of the aerosol properties to observations rather than referring to the previous paper.

Response: We have taken the reviewer's suggestion. A figure (Fig. 4a) is added for the comparison of model simulated AOD with observations from MIS (also shown below, Figure 1). We can see that the model simulation well captures the spatial distribution of AOD in California, including the maximum over the southern part of the valley area and the larger AODs over the lower lands to the southeast of the Sierra Nevada. Note that the smoother contour in MISR is due to the coarser horizontal resolution (0.5 °) of the MISR data (lines 327-331).

[Figure]

Figure 1. Spatial distribution of aerosol optical depth (AOD) averaged over October 2012 to June 2013 for (a) MISR observations, and (b) all aerosols in the CTRL simulation. 10-m wind vectors from the CTRL simulation is shown in (b).

It would be good to mark/list the observational sites that are used.

Response: Following the Reviewer's comments, the observational sites that are used are marked in Fig. 1, in which 991 DWR sites are represented by black dots; 138 CIMIS stations are represented by red dots; 32 SNOTEL sites are represented by magenta dots. The figure is also shown in the following Figure 2.

[Figure]

Figure 2. Model domain and terrain height (m). 991 DWR sites are represented by black dots; 138 CIMIS stations are represented by red dots; 32 SNOTEL sites are represented by magenta dots.

It mentions that there is no effect of aerosol upon ice in the model - can you discuss the potential impact of this? E.g., more aerosol might lead to more ice nucleating particles, which could affect snowfall/ice production, etc. Perhaps a sensitivity test could be done whereby the number of ice nucleating particles (INP) are enhanced. Is an INP scheme used, and if so which one?

Response: In the current WRF-Chem model, the aerosol effect on ice clouds is not included. ACI associated with ice clouds are more complex than that with liquid clouds. For example, a few studies have shown that negative Twomey effects may occur with aerosols and ice clouds, in which increased aerosols (and thus ice nuclei) lead to enhanced heterogeneous nucleation that is associated with larger and fewer ice crystals as compared to the homogeneous nucleation counterpart (DeMott et al., 2010; Chylek et al., 2006, Zhao et al. 2018). A recent study shows that the responses of ice crystal effective radius to aerosol loadings are modulated by water vapor amount in conjunction with several other meteorological parameters. While there is a significant negative correlation between ice effective radius and aerosol loading in moist conditions, consistent with the "Twomey effect" for liquid clouds, a strong positive correlation between the two occurs in dry conditions (Zhao et al. 2018). Despite numerous studies about the impact of aerosols on ice clouds, the role of anthropogenic aerosols in ice processes, especially over polluted regions, remains a challenging scientific issue. The effect of anthropogenic aerosols on ice formation and cloud radiative properties may be a critical pathway through which anthropogenic activities affect regional climate and present the opportunities for further studies using observations and models.

Following the Reviewer's comment, we have added the above discussion about the possible influence of the INP effect in the revised manuscript (lines 568-583).

Do the precipitation rates that are quoted include ice phase precipitation or just liquid? It would be helpful to try to separate the liquid and ice phase precipitation.

Response: In this study, the precipitation rate is for the total precipitation, including both liquid and ice phases (lines 284-285). Although we can separate the liquid and ice phase precipitation in the model, there are no reliable observational dataset to validate this partition. Thus we don't discuss the liquid and ice phase precipitation separately in this study.

Is it really the case that the transported aerosol comes from East Asia rather than more local sources? E.g. there seems to be a region of high AOD in Fig. 4d close to where Los Angeles is. Since the transported aerosol seems to be one of the biggest contributors the source regions for this should be examined more carefully. Wind arrows showing the mean flow are also needed for Fig. 4 (or Fig. 1).

Response: In this study, the transported aerosols refer to aerosols transported outside of the model domain, including aerosols from East Asia and other regions. It is clarified in the revised manuscript (lines 245-246). The mean flow from the CTRL simulation is included in Fig. 4b in the revised manuscript and Figure 1 in the response.

What causes the fairly large increases in SWE NW of the mountains?

Response: ARI causes fairly large increases in SWE NW of mountains. The ARI induced surface cooling over the Sierra Nevada, although not as strong as over the central valley, leads to reduced

snowmelt and hence slight increase in SWE, opposite to the overall aerosol effect on SWE (Fig. 6b, lines 366-369).

It would be good to comment on the fact that the anth+dust+tran effects do not seem to add up to total effects – i.e., the overall combined effect seems to be greater than the sum of the parts.

Response: We agree with the reviewer that the anth+dust+tran effects do not seem to add up to the total effects. Following the reviewer's suggestion, we have added the following discussion in the revised paper (lines 224-230):

Since the model explicitly considers different sources and types of aerosols and contains the physical processes to represent various aerosol effects (ARI, ASI, and ACI), it is useful to decompose the aerosol effects based on aerosol sources/types and pathways. Note that the overall aerosols effects are not a simple sum of different aerosol sources/types, nor a linear combination ARI, ASI, and ACI effects. Differences between various simulations, however, help to identify the effect of a single source or pathway and the decomposition approach is a common practice in the experiment design of modeling studies.

**Line-by-line comments**

Abstract – you should mention the study period before you start to talk about the results.

Response: The reviewer's comment is well taken. The study period has been added in the abstract (line 50).

L37 – "snow water equivalent (SWE)," – it is never explained what is meant by this. It sounds like it is the accumulated amount of snow that has fallen to the surface expressed as mm of water equivalent. But over the time period is never given. Presumably it is over the whole study period? This should be explained more thoroughly in the text before it is used.

Response: Snow Water Equivalent (SWE) is a common snowpack measurement. It is the amount of water contained within the snowpack and can be regarded as the depth of water over unit flat surface that would theoretically result if the entire snowpack melted instantaneously.

Following the reviewer's comment, we added the definition of SWE in the revision (lines 273-275).

L238 – Does the CPC rain rate product include only rain (and not snow)? This should be mentioned for clarity.

Response: The precipitation rate is for the total precipitation, including both rainfall and snow. It is clarified in the revised manuscript (lines 284-285).

L245 – "For SWE, daily mean SWE simulations are compared with measurements collected at Snow Telemetry" – should this be daily accumulated measurements rather than a mean?

Response: Thanks. It is corrected.

L251 – "Model data are sampled onto observational sites before the comparison is conducted" – This information needs to come before the results are discussed (and put in the caption too). Does it apply to all of the observational data? Where are the observational sites? They should be listed or marked on the map, or at least some information on how many there are and their distribution, etc.

Response: Yes, it applies to all the observations used in Fig. 3. Following the reviewer's comment, this information has been moved before the results are discussed and added in the caption. The observational sites haven been added in Fig. 1 and its caption in the revised manuscript (also in Figure 2 of the response).

L258 – "Therefore, the WRF-Chem model that we employ in this study is a reliable tool for examining the impact of aerosols on the seasonal variations of precipitation and snowpack in California, especially over the Sierra Nevada"

The results show a good representation of the meteorology and precipitation/snow, but it is a bit of an extrapolation to say that this means that it can reliably be used for aerosol-cloud interactions. E.g. we don't know how well it captures the aerosol and how its interaction with clouds. Better to say that the model represents the meteorology in a realistic manner. Or move the sentence to after you have explained how WRF compares for aerosol in the next paragraph.

Response: Following the reviewer's comment, we moved this sentence to the end of this section after the evaluation of WRF-Chem AOD and snow albedo which is related to the direct effect of ASI (line 344-347).

L283 – "Transported aerosols, including dust and biological aerosols from East Asia (Creamean et al., 2013), are carried into the domain by atmospheric circulation and widely distributed, with more over the central valley due to the trapping of aerosols by the surrounding mountains (Fig. 4d)."

Is it really the case that the transported aerosol comes from East Asia rather than more local sources? E.g. there seems to be a region of high AOD in Fig. 4d close to where Los Angeles is. Since the transported aerosol seems to be one of the biggest contributors the source regions for this should be examined more carefully.

Response: The transported aerosols refer to all aerosols transported from outside of the model domain, not just from East Asia. It is clarified in the revised manuscript (lines 245-246).

Also, can you explain how you made these plots? E.g. are they from runs with just the particular emissions included (anth, dust, trans), or did you have to do some differencing between the CTRL case and the e.g. no transport simulation?

Response: We use the difference between the CTRL simulation and the corresponding experiment (NoLocAnth, NoLocDust and NoTran), respectively, to represent the simulated AOD for local anthropogenic aerosols, local dust aerosols, or transported aerosols. It is clarified in the revised manuscript (lines 324-327).

L305 – you don't talk about the effect on SWE here even though it appears stronger than for the ARI where you did discuss it.

Response: It is discussed as follows.

The main effect of ASI is to increase the temperature (Fig. 7c) over the snowy area of the Sierra Nevada through the reduction of snow albedo (Fig. 7d) and hence more absorption of solar radiation at the surface, contributing to the reduced SWE over the Sierra (Fig. 7b) (lines 369-373).

L318 - can you elaborate on why there is less SWE due to ACIs? What is the proposed mechanism and do you have evidence for it? Is it related to their being less liquid precipitation (e.g. less raindrop freezing, smaller droplets and so less droplet freezing)? Or does precipitation here include that from snow/ice? It might be argued that the higher LWPs might allow more liquid water to become frozen giving more SWE. Later on (L408) you say that the extra clouds from the ACI effect lead to less surface melt and more SWE for the lower elevation regions – can you explain/show whether the precipitation (or other) effect dominate over the temperature effect for the mountain tops, but not the lower elevations?

 Likewise, can you please elaborate on why the albedo decreases and why the surface temperature increases. Is it due to the lack of fresh snow so that there is more exposed aged snow (although , or perhaps there are regions with no snow at all (at the start of the season perhaps)?

Response: In this study, precipitation includes rainfall, snow, and ice. Generally, precipitation increases with elevation due to orographic forcing and hence most precipitation occurs on the mountain range. Due to ACI, precipitation (including snow) over mountain range decreases, leading to reduced SWE over a large area of the Sierra Nevada. Surface snow albedo is proportional to the amount of snow on the ground. When SWE reduces, snow albedo decreases and hence the surface reflects less but absorb more solar radiation, resulting in warmer surface temperature over mountain tops.

For lower elevations, combined effect of ACI and ARI helps to cool the surface and result in less snowmelt.

L343 – "It is shown that transported aerosols also reduce the precipitation through ACI (Fig. 12a),"

Response: We are not sure what this question is about.

L432 – "the impact of aerosols is to speed up snowmelt at mountain tops." – This sentence should be removed since it suggests that aerosol enhance overall snowmelt when actually they reduce the

runoff overall. There is a small effect of speeding up the onset, but this has already been mentioned and does not need to be said again since it ignores the snowmelt reduction effect (through the precipitation decrease).

Response: Following the reviewer's comments, we rephrase the text to better explain this (lines 496-508):

For lower elevations where there is not much snow, surface runoff is mainly associated with precipitation and the changes present a similar pattern to those in precipitation (Fig. 17c). Changes in surface runoff for the whole area present similar patterns to those of the lower elevations because of the larger area of lower elevations (Fig. 17a). However for mountain tops, changes in surface runoff are also associated with changes in snowmelt. Surface runoff over mountain tops shows a slight increase in spring, and then a decrease after April (Fig. 17b). The increase can be explained by the effect of dust aerosols deposited on the snow, which reduces the snow albedo through ASI and warms the surface, leading to more and earlier snowmelt than normal, consistent with negative changes in SWE. The decrease after April is a combined effect of less snowpack available for melting caused by earlier snowmelt due to dust aerosols and reduced precipitation caused by transported and anthropogenic aerosols through ACI. Thus, the impact of aerosols is to speed up snowmelt at mountain tops in spring and modify the seasonal cycle of surface runoff.

Conclusions/L441 – "Temperature: Dust aerosols warm the mountain top surfaces through ASI (0.12 K)," – would be good to say that the numbers in brackets are domain mean changes. Also, you should reiterated the abbreviations ASI, etc. in the text at the start of the conclusions and refer to Table 4.

Response: Following the reviewer's comment, the abbreviations ARI, ASI , and ACI have been reiterated, and a brief clarification for the numbers in the brackets have been given and referred to Table 4 (lines 515-516).

L468 – "Therefore, one of the important impacts of aerosols is to speed up the snowmelt at mountain tops." Is this really one of the most important aspects? Since the effect on runoff then goes on to be dominated by the reduction in the precipitation. And you can't be sure how much effect the earlier snow melt is having on that – most of the effect could be coming from the precip reduction?

Response: We agree with the reviewer that changes in runoff are dominated by changes in the precipitation. However, snowmelt also plays an important role in warm and dry season (lines 495-508). The earlier snowmelt at mountain tops induced by aerosols is important for water management since California depends heavily on snowmelt for water use in dry seasons.

**Tables/Figures**

Table 3 – perhaps it is worth mentioning that these experiments use the CTRL aerosol emissions.

Response: Done (Table 3).

Fig. 1 – It would be useful to label the valley, big cities and other regions of interest in Fig. 1. Also, the colorbar is a bit strange since the colors around 150m and 600m seem to repeat.

Response: Following the reviewer's suggestion, the indicators for mountains and big cities have been provided in Fig. 1. The colorbar in Fig. 1 is also changed. It is shown in Figure 2 of the response.

Fig. 2 – it is confusing to say that the SWE is averaged over the time period since presumably it is the accumulated snow amount?

Response: Here the model simulated SWE is the mean value of the accumulated SWE from 3-hourly model outputs. It is clarified in the revised manuscript (lines 276-277).

Fig.3 – should state the region being considered here and in the text – is it the whole model domain? It would be good to also use a dashed line for the model to help distinguish it for colorblind readers.

Response: It is the mean values at the corresponding observational sites. It is clarified in the caption. Sites are identified in Fig. 1 in the revised manuscript. Dashed line is used for the model results as the reviewer suggested.

**Typos**

L230 – "in CTRL experiment" -> "in the CTRL experiment"

Response: Corrected (line 272).

L233 - "in the northern California" -> "in northern California"

Response: Corrected (line 279).

L235 – "while colder temperature is found" -> "while colder temperatures are found"

Response: Corrected (line 281).

L314 - "because aerosol effect" -> "because the aerosol effect"

Response: Corrected (line 378).

L316 - "associated with ACI effect" -> "associated with the ACI effect"

Response: Corrected (line 381).

L358 – "contributes to the increase (1.88%)." – "contributes to an increase (1.88%)." (since overall there is a decrease).

L484 – "importance" -> "important"